# Provably Robust Boosted Decision Stumps and Trees against Adversarial Attacks

**Maksym Andriushchenko**
University of Tübingen
maksym.andriushchenko@uni-tuebingen.de

**Matthias Hein**
University of Tübingen
matthias.hein@uni-tuebingen.de

## Abstract

The problem of adversarial robustness has been studied extensively for neural networks. However, for boosted decision trees and decision stumps there are almost no results, even though they are widely used in practice (e.g. XGBoost) due to their accuracy, interpretability, and efficiency. We show in this paper that for boosted decision stumps the *exact* min-max robust loss and test error for an $l_\infty$-attack can be computed in $O(T \log T)$ time per input, where $T$ is the number of decision stumps and the optimal update step of the ensemble can be done in $O(n^2 \, T \log T)$, where $n$ is the number of data points. For boosted trees we show how to efficiently calculate and optimize an upper bound on the robust loss, which leads to state-of-the-art robust test error for boosted trees on MNIST (12.5% for $\epsilon_\infty = 0.3$), FMNIST (23.2% for $\epsilon_\infty = 0.1$), and CIFAR-10 (74.7% for $\epsilon_\infty = 8/255$). Moreover, the robust test error rates we achieve are competitive to the ones of provably robust convolutional networks. The code of all our experiments is available at http://github.com/max-andr/provably-robust-boosting.

## 1 Introduction

It has recently been shown that deep neural networks are easily fooled by imperceptible perturbations called *adversarial examples* [62, 24] or tend to output high-confidence predictions on out-of-distribution inputs [51, 49, 29] that have nothing to do with the original classes. The most popular defense against adversarial examples is adversarial training [24, 45], which is formulated as a robust optimization problem [59, 45]. However, the inner maximization problem is likely to be NP-hard for neural networks as computing optimal adversarial examples is NP-hard [33, 71]. A large variety of sophisticated defenses proposed for neural networks [31, 7, 43] could be broken again via more sophisticated attacks [1, 18, 48]. Moreover, empirical robustness, evaluated by *some* attack, can also arise from gradient masking or obfuscation [1] in which case gradient-free or black-box attacks often break heuristic defenses. A solution to this problem are methods that lead to *provable robustness guarantees* [28, 72, 54, 77, 68, 75, 13, 25] or lead to classifiers which can be certified via exact combinatorial solvers [63]. However, these solvers do not scale to large neural networks, and networks having robustness guarantees lack in terms of prediction performance compared to standard ones. The only scalable certification method is randomized smoothing [41, 42, 12, 57], however obtaining tight certificates for norms other than $l_2$ is an open research question.

While the adversarial problem has been studied extensively for neural networks, other classifiers have received much less attention e.g. kernel machines [76, 56, 28], k-nearest neighbors [69], and decision trees [52, 3, 9]. Boosting, in particular boosted decision trees, are very popular in practice due to their interpretability, competitive prediction performance, and efficient recent implementations such as XGBoost [10] and LightGBM [34]. Thus there is also a need to develop boosting methods which are robust to worst-case measurement error or adversarial changes of the input data. While robust boosting has been extensively considered in the literature [70, 44, 19], it refers in that context to a



**Figure 1: Left**: boosted decision *stumps*: normal and our robust models. **Right**: boosted decision *trees*: normal and our robust models. In both cases, the normal models have very small geometric margin, while our robust models also classify all training points correctly but additionally enforce a large geometric margin.

large functional margin or robustness with respect to outliers e.g. via using a robust loss function, but not to the adversarial robustness we are considering in this paper. In the context of *adversarial* robustness, very recently [9] considered the robust min-max loss for an ensemble of decision trees with coordinate-aligned splits. They proposed an approximation of the inner maximization problem but without any guarantees. The robustness guarantees were then obtained via a mixed-integer formulation of [32] for the computation of the minimal adversarial perturbation for tree ensembles. However, this approach has limited scalability to large problems.

**Contributions** In this paper, we show how to exactly compute the robust loss and robust test error with respect to $l_\infty$-norm perturbations for an ensemble of decision stumps with coordinate-aligned splits. This can be done efficiently in $O(T \log T)$ time per data point, where $T$ is the number of decision stumps. Moreover, we show how to perform the globally optimal update of an ensemble of decision stumps by directly minimizing the robust loss without any approximation in $O(n^2\, T \log T)$ time per coordinate, where $n$ is the number of training examples. We also derive a strict upper bound on the robust loss for tree ensembles based on our results for an ensemble of decision stumps. It can be efficiently evaluated in $O(T\, l)$ time, where $l$ is the number of leaves in the tree. Then we show how this upper bound can be minimized during training in $O(n^2\, l)$ time per coordinate. Our derived upper bound is quite tight empirically and leads to provable guarantees on the robustness of the resulting tree ensemble. The difference of the resulting robust boosted decision stumps and trees compared to normally trained models is visualized in Figure 1.

## 2 Boosting and Robust Optimization for Adversarial Robustness

In this section we fix the notation, the framework of boosting, and define briefly the basis of robust optimization for adversarial robustness, underlying adversarial training. In the next sections we derive the specific robust training procedure for an ensemble of decision stumps where we optimize the exact robust loss and for a tree ensemble where we optimize an upper bound.

**Boosting** While the main ideas can be generalized to the multi-class setting (using one-vs-all, see Appendix E), for simplicity of the derivations we restrict ourselves to binary classification, that is our labels $y$ are in $\{-1, 1\}$ and we assume to have $d$ real-valued features. Boosting can be described as the task of fitting an ensemble $F : \mathbb{R}^d \to \mathbb{R}$ of weak learners $f_t : \mathbb{R}^d \to \mathbb{R}$ given as $F(x) = \sum_{t=1}^{T} f_t(x)$. The final classification is done via the sign of $F(x)$. In boosting the ensemble is fitted in a greedy way in the sense that given the already estimated ensemble we determine an update $F' = F + f_{T+1}$, by fitting the new weak learner $f_{T+1}$ being guided by the performance of the current ensemble $F$. In this paper we use in the experiments the exponential loss $L : \mathbb{R} \to \mathbb{R}$, where we use the functional margin formulation where for a point $(x, y) \in \mathbb{R}^d \times \{-1, 1\}$ it is defined as $L(y\, f(x)) = \exp(-y\, f(x))$. However, all following algorithms and derivations hold for any margin-based, strictly monotonically decreasing, convex loss function $L$, e.g. logistic loss $L(y\, f(x)) = \ln(1 + \exp(-yf(x)))$. The advantage of the exponential loss is that it decouples $F$ and the update $f_{T+1}$ in the estimation process and allows us to see the estimation process for $f_{T+1}$ as fitting a weighted exponential loss where the weights to fit $(x, y)$ are given by $\exp(-y\, F(x))$,

$$L(y\, F'(x)) = \exp\big(-y\,\big(F(x) + f_{T+1}(x)\big)\big) = \exp\big(-y\, F(x)\big) \exp\big(-y\, f_{T+1}(x)\big).$$

In this paper we consider as weak learners: a) decision stumps (i.e. trees of depth one) of the form $f_{t,i} : \mathbb{R}^d \to \mathbb{R}$, $f_{t,i}(x) = w_l + w_r \mathbb{1}_{x_i \geq b}$, where one does a coordinate-aligned split and b) decision

trees (binary tree) of the form $f_t(x) = u_{q_t(x)}^{(t)}$, where $u_{q_t(x)}^{(t)} : V \rightarrow \mathbb{R}$ is a mapping from the set of leaves $V$ of the tree to $\mathbb{R}$ and $q_t : \mathbb{R}^d \rightarrow V$ is a mapping which assigns to every input the leaf of the tree it ends up. While the approach can be generalized to general linear splits of the form, $w_l + w_r \mathbb{1}_{\langle v, x \rangle \geq b}$, we concentrate on coordinate-aligned splits, $w_l + w_r \mathbb{1}_{x_i \geq b}$ which are more common in practice since they lead to competitive performance and are easier to interpret for humans.

**Robust optimization for adversarial robustness**  Finding the minimal perturbation with respect to some $l_p$-distance can be formulated as the following optimization problem:

$$\min_{\delta \in \mathbb{R}^d} \|\delta\|_p \quad \text{such that} \quad y_i f(x_i + \delta) \leq 0, \quad x_i + \delta \in C \tag{1}$$

where $(x_i, y_i) \in \mathbb{R}^d \times \{-1, 1\}$ and $C$ is a set of constraints every input has to fulfill. In this paper we assume $C = [0, 1]^d$ and that all features are normalized to be in this range. We emphasize that we concentrate on *continuous* features, for adversarial perturbations of *discrete* features we refer to [53, 17, 36]. We denote by $\delta_i^*$ the optimal solution of this problem for $(x_i, y_i)$. Furthermore, let $\Delta_p(\epsilon) := \{\delta \in \mathbb{R}^d \mid \|\delta\|_p \leq \epsilon\}$ be the set of perturbations with respect to which we aim to be robust. Then the *robust test error* with respect to $\Delta_p(\epsilon)$ is defined for $n$ data points as $\frac{1}{n} \sum_{i=1}^{n} \mathbb{1}_{\|\delta_i^*\|_p \leq \epsilon}$.

The optimization problem (1) is non-convex for neural networks and can only be solved exactly via mixed-integer programming [63] which scales exponentially with the number of hidden neurons. Since such an evaluation is prohibitively expensive in most cases, often robustness is evaluated via heuristic attacks [47, 45, 8] which results in lower bounds on the robust test error. Provable robustness aims at providing upper bounds on the robust test error and the optimization of these bounds during training [28, 72, 54, 77, 75, 13, 25, 12]. For an ensemble of trees the optimization problem (1) can also be reformulated as a mixed-integer-program [32] which does not scale to large ensembles.

The goal of improving adversarial robustness can be formulated as a robust optimization problem with respect to the set of allowed perturbations $\Delta_p(\epsilon)$ [59, 45]:

$$\min_{\theta} \sum_{i=1}^{n} \max_{\delta \in \Delta_p(\epsilon)} L\big(f(x_i + \delta; \theta), y_i\big). \tag{2}$$

A training process, where one tries at each update step to approximately solve the inner maximization problem, is called *adversarial training* [24]. We note that the maximization problem is in general non-concave and thus globally optimal solutions are very difficult to obtain. Our goal in the following two sections is to get provable robustness guarantees for boosted stumps and trees by directly optimizing (2) or an upper bound on the inner maximization problem.

## 3   Exact Robust Optimization for Boosted Decision Stumps

We first show how the exact robust loss $\max_{\delta \in \Delta_p(\epsilon)} L(y_i F(x_i + \delta; \theta))$ can be computed for an ensemble $F$ of decision stumps. While decision stumps are very simple weak learners, they have been used in the original AdaBoost [20] and were successfully used in object detection [66] or face detection [67] which could be done in real-time due to the simplicity of the classifier.

### 3.1   Exact Robust Test Error for Boosted Decision Stumps

The ensemble of decision stumps can be written as

$$F(x) = \sum_{t=1}^{T} f_{t, c_t}(x) = \sum_{t=1}^{T} \left( w_l^{(t)} + w_r^{(t)} \mathbb{1}_{x_{c_t} \geq b_t} \right),$$

where $c_t$ is the coordinate for which $f_t$ makes a split. First, observe that a point $x \in \mathbb{R}^d$ with label $y$ is correctly classified when $yF(x) > 0$. In order to determine whether the point $x$ is adversarially robust wrt $l_\infty$-perturbations, one has to solve the following optimization problem:

$$G(x, y) := \min_{\|\delta\|_\infty \leq \epsilon} yF(x + \delta) \tag{3}$$

If $G(x, y) \leq 0$, then the point $x$ is non-robust. If $G(x, y) > 0$, then the point $x$ is robust, i.e. it is not possible to change the class. Thus the exact minimization of (3) over the test set yields the exact robust test error. For many state-of-the-art classifiers, this problem is NP-hard. For particular MIP formulations for tree ensembles, see [32], or for neural networks, see [63]. Closed-form solutions are known only for the simplest models such as linear classifiers [24].

We can solve this certification problem for the robust test error exactly and efficiently by noting that the objective and the attack model $\Delta_\infty(\epsilon)$ is *separable* wrt the input dimensions. Therefore, we have to solve up to $d$ simple *one-dimensional* optimization problems. We denote $S_k = \{s \in \{1, \ldots, T\} \mid c_s = k\}$, i.e. the set of stump indices that split coordinate $k$. Then

$$\min_{\|\delta\|_\infty \leq \epsilon} yF(x + \delta) = \min_{\|\delta\|_\infty \leq \epsilon} \sum_{t=1}^{T} yf_{t,c_t}(x + \delta) = \min_{\|\delta\|_\infty \leq \epsilon} \sum_{k=1}^{d} \sum_{s \in S_k} yf_{s,k}(x + \delta) \tag{4}$$

$$= \sum_{k=1}^{d} \min_{|\delta_k| \leq \epsilon} \sum_{s \in S_k} yf_{s,k}(x + \delta) = \sum_{k=1}^{d} \Big[ \sum_{s \in S_k} yw_l^{(s)} + \min_{|\delta_k| \leq \epsilon} \sum_{s \in S_k} yw_r^{(s)} \mathbb{1}_{x_k + \delta_k \geq b_s} \Big] := \sum_{k=1}^{d} G_k(x, y)$$

The one-dimensional optimization problem $\min_{|\delta_k| \leq \epsilon} \sum_{s \in S_k} yw_r^{(s)} \mathbb{1}_{x_k + \delta_k \geq b_s}$ can be solved by simply checking all $|S_k| + 1$ piece-wise constant regions of the classifier for $\delta_k \in [-\epsilon, \epsilon]$. The detailed algorithm can be found in Appendix B. The overall time complexity of the exact certification is $O(T \log T)$ since we need to sort up to $T$ thresholds $b_s$ in ascending order to efficiently calculate all partial sums of the objective. Moreover, using this result, we can obtain provably minimal adversarial examples (see Appendix B for details and Figure 11 for visualizations).

## 3.2 Exact Robust Loss Minimization for Boosted Decision Stumps

We note that when $L$ is monotonically decreasing, it holds:

$$\max_{\delta \in \Delta_\infty(\epsilon)} L(y\, F(x + \delta)) = L\Big( \min_{\delta \in \Delta_\infty(\epsilon)} yF(x + \delta) \Big),$$

and thus the certification algorithm can directly be used to compute also the robust loss. For updating the ensemble $F$ with a new stump $f$ that splits a certain coordinate $j$, we first have to solve the inner maximization problem over $\Delta_\infty(\epsilon)$ in (2) before[1] we optimize the parameters $w_l, w_r, b$ of $f$:

$$\max_{\|\delta\|_\infty \leq \epsilon} L\Big( y_i F(x_i + \delta) + y_i f_j(x_i + \delta) \Big) = L\Big( \min_{\|\delta\|_\infty \leq \epsilon} \Big[ \sum_{k=1}^{d} \sum_{s \in S_k} y_i f_{s,k}(x_i + \delta) + y_i f_j(x_i + \delta) \Big] \Big)$$

$$= L\Big( \sum_{k \neq j} \min_{|\delta_k| \leq \epsilon} \sum_{s \in S_k} y_i f_{s,k}(x_i + \delta) + \min_{|\delta_j| \leq \epsilon} \Big[ \sum_{s \in S_j} y_i f_{s,j}(x_i + \delta) + y_i f_j(x_i + \delta) \Big] \Big)$$

$$= L\Big( \sum_{k \neq j} G_k(x_i, y_i) + \sum_{s \in S_j} y_i w_l^{(s)} + y_i w_l + \min_{|\delta_j| \leq \epsilon} \Big[ \sum_{s \in S_j} y_i w_r^{(s)} \mathbb{1}_{x_{ij} + \delta_j \geq b_s} + y_i w_r \mathbb{1}_{x_{ij} + \delta_j \geq b} \Big] \Big).$$

In order to solve the remaining optimization problem for $\delta_j$ we have to make a case distinction based on the values of $w_r$. However, first we define the minimal values of the ensemble part on $\delta_j \in [-\epsilon, b - x_{ij})$ and $\delta_j \in [b - x_{ij}, \epsilon]$ as

$$h_l(x_{ij}, y_i) := \min_{\substack{\delta_j < b - x_{ij} \\ |\delta_j| \leq \epsilon}} \sum_{s \in S_j} y_i w_r^{(s)} \mathbb{1}_{x_{ij} + \delta_j \geq b_s}, \quad h_r(x_{ij}, y_i) := \min_{\substack{\delta_j \geq b - x_{ij} \\ |\delta_j| \leq \epsilon}} \sum_{s \in S_j} y_i w_r^{(s)} \mathbb{1}_{x_{ij} + \delta_j \geq b_s}$$

These problems can be solved analogously to $G_k(x, y)$. Then we get the case distinction:

$$g(x_{ij}, y_i; w_r) = \min_{|\delta_j| \leq \epsilon} \Big[ \sum_{s \in S_j} y_i w_r^{(s)} \mathbb{1}_{x_{ij} + \delta_j \geq b_s} + y_i w_r \mathbb{1}_{x_{ij} + \delta_j \geq b} \Big] \tag{5}$$

$$= \begin{cases} h_r(x_{ij}, y_i) + y_i w_r & \text{if } b - x_{ij} < -\epsilon \text{ or } (|b - x_{ij}| \leq \epsilon \text{ and } h_l(x_{ij}, y_i) > h_r(x_{ij}, y_i) + y_i w_r) \\ h_l(x_{ij}, y_i) & \text{if } b - x_{ij} > \epsilon \quad \text{or } (|b - x_{ij}| \leq \epsilon \text{ and } h_l(x_{ij}, y_i) \leq h_r(x_{ij}, y_i) + y_i w_r) \end{cases}$$

The following Lemma shows that the robust loss is jointly convex in $w_l, w_r$ (to see this set $l = 2$, $u = (w_l, w_r)^T$, $r(\hat{x}) = (y_i, y_i \mathbb{1}_{\hat{x}_{ij} \geq b})^T$, $C = B_\infty(x_i, \epsilon)$ and $c = \sum_{k \neq j} G_k(x_i, y_i)$).

**Lemma 1** *Let $L : \mathbb{R} \to \mathbb{R}$ be a convex, monotonically decreasing function. Then $\tilde{L} : \mathbb{R}^l \to \mathbb{R}$ defined as $\tilde{L}(u) = \max_{\tilde{x} \in C} L(c + \langle r(\tilde{x}), u \rangle)$ is convex for any $c \in \mathbb{R}$, $r : \mathbb{R}^d \to \mathbb{R}^l$, and $C \subseteq \mathbb{R}^d$.*

Thus the loss term for each data point is jointly convex in $w_l, w_r$ and consequently the sum of the losses is convex as well. This means that for the overall robust optimization problem over the parameters $w_l, w_r$ (for a fixed $b$), we have to minimize the following convex function

$$L^*(j,b) = \min_{w_l, w_r} \sum_{i=1}^{n} L\Big( \sum_{k \neq j} G_k(x_i, y_i) + \sum_{s \in S_j} y_i w_l^{(s)} + y_i w_l + g(x_{ij}, y_i; w_r) \Big).$$

We plot an example of this objective wrt the parameters $w_l$ and $w_r$ of a single decision stump in Figure 2. In general, for an arbitrary loss $L$, there is no closed-form minimizer wrt $w_l$ and $w_r$. Thus, we can minimize such an objective using, e.g. coordinate descent. Then on every iteration of coordinate descent the minimum wrt $w_l$ or $w_r$ can be found using bisection for any convex loss $L$. For the exponential loss, we can optimize wrt $w_l$ via a closed-form minimizer when $w_r$ is fixed. The details can be found in Appendix B.3.

Finally, we have to minimize over all possible thresholds. We choose the potential thresholds $b \in B_j = \{x_{ij} - \epsilon - \nu, x_{ij} + \epsilon + \nu \mid i = 1, \ldots, n\}$, where $\nu$ can be as small as precision allows and is just introduced so that the thresholds lie outside of $\Delta_\infty(\epsilon)$. We optimize the robust loss $L^*(j, b)$ for all thresholds $b \in B_j$ and determine the minimum. For each contiguous set of minimizers we determine the nearest neighbors in $B_j$ and check the thresholds half-way to them (note that they have at most the same robust loss but never a better one) and then take the threshold in the middle of all the ones having equal loss. As there are in the worst case $2n$ unique thresholds, the overall complexity of one update step is $O(n^2 T \log T)$. And finally, at each update step one typically checks all $d$ coordinates and takes the one which yields the smallest overall robust loss of the ensemble.

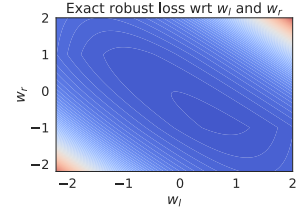

**Figure 2:** Visualization of the min-max objective which is convex wrt the parameters $w_l$ and $w_r$ of a decision stump.

## 4   Robust Optimization for Boosted Decision Trees

We first provide an upper bound on the robust test error of the tree ensemble which is used further to derive an upper bound on the robust loss that is then minimized in the update step of tree ensemble.

### 4.1   Upper Bound on the Robust Test Error for Boosted Decision Trees

Our goal is to solve the optimization problem (3). While the exact minimization is NP-hard for tree ensembles [32], we can similarly to [73, 54] for neural networks derive a tractable lower bound $\tilde{G}(x, y)$ on $G(x, y)$ for an ensemble of trees:

$$\min_{\|\delta\|_p \leq \epsilon} yF(x+\delta) = \min_{\|\delta\|_p \leq \epsilon} \sum_{t=1}^{T} yu_{q_t(x+\delta)}^{(t)} \geq \sum_{t=1}^{T} \min_{\|\delta\|_p \leq \epsilon} yu_{q_t(x+\delta)}^{(t)} := \tilde{G}(x, y) \qquad (6)$$

If $\tilde{G}(x, y) > 0$, then the point $x$ is provably robust. However, if $\tilde{G}(x, y) \leq 0$, the point may be either robust or non-robust. In this way, we get an upper bound on the number of non-robust points, which yields an *upper bound* on the robust test error. We note that for a decision tree, $\min_{\|\delta\|_p \leq \epsilon} yu_{q_t(x+\delta)}^{(t)}$ can be found exactly by checking all leafs which are reachable for points in $B_p(x, \epsilon)$. This can be done in $O(l)$ time per tree, where $l$ is the number of leaves in the tree.

### 4.2   Minimization of an Upper Bound on the Robust Loss for Boosted Decision Trees

The goal is to upper bound the inner maximization problem of Equation (2) based on the certificate that we derived. Note that we aim to *bound* the loss of the whole ensemble $F + f$, and thus we do not use any approximations of the loss such as the second-order Taylor expansion used in [23, 10]. We use $p = \infty$, that is the attack model is $\Delta_\infty(\epsilon)$. Let $F(x) = \sum_{t=1}^{T} f_t(x) = \sum_{t=1}^{T} u_{q_t(x)}^{(t)}$ be a fixed

ensemble of trees and $f$ a new tree with which we update the ensemble. Then the robust optimization problem is:

$$\min_f \sum_{i=1}^n \max_{\|\delta\|_\infty \leq \epsilon} L\Big(y_i\big(F(x_i + \delta) + f(x_i + \delta)\big)\Big) \tag{7}$$

The inner maximization problem can be upper bounded for every tree separately given that $L(yf(x))$ is monotonically decreasing wrt $yf(x)$, and using our certificate for the ensemble of $T + 1$ trees:

$$\max_{\|\delta\|_\infty \leq \epsilon} L\Big(y_i F(x_i + \delta) + y_i f(x_i + \delta)\Big) = L\Big(\min_{\|\delta\|_\infty \leq \epsilon} \Big[\sum_{t=1}^T y_i f_t(x_i + \delta) + y_i f(x_i + \delta)\Big]\Big) \tag{8}$$

$$\leq L\Big(\sum_{t=1}^T \min_{\|\delta\|_\infty \leq \epsilon} y_i f_t(x_i + \delta) + \min_{\|\delta\|_\infty \leq \epsilon} y_i f(x_i + \delta)\Big) = L\Big(\tilde{G}(x_i, y_i) + \min_{\|\delta\|_\infty \leq \epsilon} y_i f(x_i + \delta)\Big)$$

We can efficiently calculate $\tilde{G}(x_i, y_i)$ as described in the previous subsection. But note that $\min_{\|\delta\|_\infty \leq \epsilon} y_i f(x_i + \delta)$ depends on the tree $f$. The exact tree fitting is known to be NP-complete [39], although it is still possible to scale it to some moderate-sized problems with recent advances in MIP-solvers and hardware as shown in [2]. We want to keep the overall procedure scalable to large datasets, so we will stick to the standard greedy recursive algorithm for fitting the tree. On every step of this process, we fit for some coordinate $j \in \{1, \ldots, d\}$ and for some splitting threshold $b$, a single decision stump $f(x) = w_l + w_r \mathbb{1}_{x_j \geq b}$. Therefore, for a particular decision stump with threshold $b$ and coordinate $j$ we have to solve the following problem:

$$\min_{w_l, w_r \in \mathbb{R}} \sum_{i \in I} L\Big(\tilde{G}(x_i, y_i) + y_i w_l + \min_{|\delta_j| \leq \epsilon} y_i w_r \mathbb{1}_{x_{ij} + \delta_j \geq b}\Big) \tag{9}$$

where $I$ are all the points $x_i + \delta$ which can reach this leaf for some $\delta$ with $\|\delta\|_\infty \leq \epsilon$.

Finally, we have to make a case distinction depending on the values of $w_r$ and $b - x_{ij}$:

$$\min_{|\delta_j| \leq \epsilon} y_i w_r \mathbb{1}_{x_{ij} + \delta_j \geq b} = y_i w_r \cdot \begin{cases} 1 & \text{if } b - x_{ij} < -\epsilon \text{ or } (|b - x_{ij}| \leq \epsilon \text{ and } y_i w_r < 0) \\ 0 & \text{if } b - x_{ij} > \epsilon \quad \text{or } (|b - x_{ij}| \leq \epsilon \text{ and } y_i w_r \geq 0) \end{cases} \tag{10}$$

where we denote the case distinction for brevity as $\mathbb{1}(x_i, y_i; w_r)$. Note that the right side of (10) is concave as a function of $w_r$. Thus the overall robust optimization amounts to finding the minimum of the following objective, which is again by Lemma 1 jointly convex in $w_l, w_r$:

$$L^*(j, b) = \min_{w_l, w_r} \sum_{i : i \in I} L\Big(\tilde{G}(x_i, y_i) + y_i w_l + y_i w_r \mathbb{1}(x_i, y_i; w_r)\Big) \tag{11}$$

Note that the case distinction $\mathbb{1}(x_i, y_i; w_r)$ can be fixed once we fix the sign of $w_r$. This allows us to avoid doing bisection on $w_r$, and rather use coordinate descent directly on each interval $w_r \geq 0$ and $w_r < 0$. After finding the minimum of the objective on each interval, we then combine the results from both intervals by taking the smallest loss out of them. The details are given in Appendix B.3.

Then we select the optimal threshold as described in Section 3.2. Finally, as in other tree building methods such as [5, 10], we perform pruning after a tree is constructed. We start from the leafs and prune nodes based on the upper bound on the training robust loss (8) to ensure that it decreases at every iteration of tree boosting. This cannot be guaranteed with robust splits without pruning since the tree construction process is greedy, and some training examples are also influenced by splits at different branches. Thus, in order to control the upper bound on the robust loss globally over the whole tree as in (8), and not just for the current subtree as in (9), we need a post-hoc approach that takes into account the structure of the whole tree. Therefore, we have to use pruning. We note that in the extreme case, pruning may leave only one decision stump at the root (although it happens extremely rarely in practice), for which we are guaranteed to decrease the upper bound on the robust loss. Thus every new tree in the ensemble is guaranteed to reduce the upper bound on the robust loss. Note that this is also true if we use the shrinkage parameter [21] which we discuss in Appendix C.

Lastly, we note that the total worst case complexity is $O(n^2)$ in the number of training examples compared to $O(n \log n)$ for XGBoost, which is a relatively low price given that the overall optimization problem is significantly more complicated than the formulation used in XGBoost.

# 5 Experiments

**General setup** We are primarily interested in two quantities: test error (TE) and robust test error (RTE) wrt $l_\infty$-perturbations. For boosted stumps, we compute RTE as described in Section 3.1, but we also report the upper bound on RTE (URTE) obtained using the stump-wise bound from Section 4.1 to illustrate that it is actually tight for almost all models. For boosted trees, we report RTE obtained via the MIP formulation of [32] which we adapted to a feasibility problem (see Appendix G.2 for more details), and also the tree-wise upper bounds described in Section 4.1. For evaluation we use 11 datasets: breast-cancer, diabetes, cod-rna, MNIST 1-5 (digit 1 vs digit 5), MNIST 2-6 (digit 2 vs digit 6, following [32] and [9]), FMNIST shoes (sandals vs sneakers), GTS 100-rw (speed 100 vs roadworks), GTS 30-70 (speed 30 vs speed 70), MNIST, FMNIST, and CIFAR-10. More details about the datasets are given in Appendix F. We emphasize that we evaluate our models on image recognition datasets mainly for the sake of comparison to other methods reported in the literature.

We consider five types of boosted stumps: normally trained stumps, adversarially trained stumps (see Appendix G.1 for these results), robust stumps of Chen et al. [9], our robust stumps where the robust loss is bounded stump-wise, and our robust stumps where the robust loss is calculated exactly. Next we consider four types of boosted trees: normally trained trees, adversarially trained trees, robust trees of Chen et al. [9], and our robust trees where the robust loss is bounded tree-wise. Both for stumps and trees, we perform $l_\infty$ adversarial training following [32], i.e. every iteration we train on clean training points and adversarial examples (equal proportion). We generate adversarial examples via the *cube attack* – a simple attack inspired by random search [50] described in Appendix D (we use 10 iterations and $p = 0.5$) and its performance is shown in Section G.3. We perform model selection of our models and models from Chen et al. [9] based on the validation set of 20% randomly selected points from the original training set, and we train on the rest of the training set. All models are trained with the exponential loss. More details about the experiments are available in Appendix F and in our repository `http://github.com/max-andr/provably-robust-boosting`.

**Boosted decision stumps** The results for boosted stumps are given in Table 1. First, we observe that normal models are not robust for the considered $l_\infty$-perturbations. However, both variants of our robust boosted stumps significantly improve RTE, outperforming the method of Chen et al. [9] on 7 out of the 8 datasets. Note that although our exact method optimizes the exact robust loss, we are still not guaranteed to *always* outperform Chen et al. [9] since they use a different loss function, and the quantities of interest are calculated on test data. The largest improvements compared to normal models are obtained on breast-cancer from 98.5% RTE to 10.9% and on MNIST 2-6 from 99.9% to 9.1% RTE. The robust models perform slightly worse in terms of test error, which is in line with the empirical observation made for adversarial training for neural networks [64]. Additionally, to the robust test error (RTE), we also report the upper bound (URTE) to show that it is very close to RTE. Notably, for our robust stumps trained with the upper bound on the robust loss, URTE is equal to the RTE for all models, and it is very close to the RTE of our robust stumps trained with the exact robust loss, while taking about 4x less time to train in average. Thus bounding the sum over weak learners element-wise, as done in (6), seems to be tight enough to yield robust models. Finally, we provide in Appendix G.2 a more detailed comparison to the robust boosted stumps of Chen et al. [9].

**Table 1:** Evaluation of robustness for boosted stumps. We show, in percentage, test error (TE), exact robust test error (RTE), and upper bound on robust test error (URTE). Both variants of our robust stumps outperform the method of Chen et al. [9]. We also observe that URTE is very close to RTE or even the same for many models.

| Dataset | $l_\infty\ \epsilon$ | Normal stumps (standard training) | | | Robust stumps Chen et al. [9] | | Our robust stumps (robust loss bound) | | | Our robust stumps (exact robust loss) | | |
|---|---|---|---|---|---|---|---|---|---|---|---|---|
| | | TE | RTE | URTE | TE | RTE | TE | RTE | URTE | TE | RTE | URTE |
| breast-cancer | 0.3 | **2.9** | 98.5 | 100 | 8.8 | 16.8 | 4.4 | **10.9** | 10.9 | 5.1 | **10.9** | 10.9 |
| diabetes | 0.05 | 24.7 | 54.5 | 56.5 | **23.4** | **30.5** | 28.6 | 33.1 | 33.1 | 27.3 | 31.8 | 31.8 |
| cod-rna | 0.025 | **4.7** | 42.8 | 44.9 | 11.6 | 23.2 | 11.2 | **22.4** | 22.4 | 11.2 | 22.6 | 22.6 |
| MNIST 1-5 | 0.3 | **0.5** | 85.4 | 85.4 | 0.9 | 5.2 | 0.6 | 3.7 | 3.7 | 0.7 | **3.6** | 3.7 |
| MNIST 2-6 | 0.3 | **1.7** | 99.9 | 99.9 | 2.8 | 13.9 | 3.0 | **9.1** | 9.1 | 3.0 | 9.2 | 9.2 |
| FMNIST shoes | 0.1 | **2.4** | 100 | 100 | 7.1 | 22.2 | 6.2 | 11.8 | 11.8 | 5.7 | **10.8** | 11.5 |
| GTS 100-rw | 8/255 | **1.1** | 9.9 | 9.9 | 2.0 | 11.8 | 2.8 | 8.9 | 8.9 | 2.0 | **6.7** | 6.7 |
| GTS 30-70 | 8/255 | **11.3** | 53.7 | 53.7 | 12.7 | 28.2 | 12.7 | **26.9** | 26.9 | 12.9 | 27.6 | 27.6 |

**Table 2:** Evaluation of robustness for boosted *trees*. We report, in percentages, test error (TE), robust test error (RTE), and upper bound on robust test error (URTE). Our robust boosted trees lead to better RTE compared to adversarial training and robust trees of Chen et al. [9]. We observe that especially for our models URTE are very close to RTE, while URTE are orders of magnitude faster to compute.

| Dataset | $l_\infty\ \epsilon$ | Normal trees (standard training) | | | Adv. trained trees (with cube attack) | | | Robust trees Chen et al. [9] | | Our robust trees (robust loss bound) | | |
|---|---|---|---|---|---|---|---|---|---|---|---|---|
| | | TE | RTE | URTE | TE | RTE | URTE | TE | RTE | TE | RTE | URTE |
| breast-cancer | 0.3 | 0.7 | 81.0 | 81.8 | **0.0** | 27.0 | 27.0 | 0.7 | 13.1 | 0.7 | **6.6** | 6.6 |
| diabetes | 0.05 | 22.7 | 55.2 | 61.7 | 26.6 | 46.8 | 46.8 | **22.1** | 40.3 | 27.3 | **35.7** | 35.7 |
| cod-rna | 0.025 | **3.4** | 37.6 | 47.1 | 10.9 | 24.8 | 24.8 | 10.2 | 24.2 | 6.9 | **21.3** | 21.4 |
| MNIST 1-5 | 0.3 | **0.1** | 90.7 | 96.0 | 1.3 | 9.0 | 9.5 | 0.3 | 2.9 | 0.2 | **1.3** | 1.4 |
| MNIST 2-6 | 0.3 | **0.4** | 89.6 | 100 | 2.3 | 15.1 | 15.9 | 0.5 | 6.9 | 0.7 | **3.8** | 4.1 |
| FMNIST shoes | 0.1 | **1.7** | 99.8 | 99.9 | 5.5 | 14.1 | 14.2 | 3.1 | 13.2 | 3.6 | **8.0** | 8.1 |
| GTS 100-rw | 8/255 | **0.9** | 6.0 | 6.1 | 1.0 | 8.4 | 8.4 | 1.5 | 9.7 | 2.6 | **4.7** | 4.7 |
| GTS 30-70 | 8/255 | 14.2 | 31.4 | 32.6 | 16.2 | 26.7 | 26.8 | **11.5** | 28.8 | 13.8 | **20.9** | 21.4 |

**Boosted decision trees** The results for boosted trees of depth 4 are given in Table 2. Our robust training of boosted trees outperforms both adversarial training and the method of Chen et al. [9] in terms of RTE on all 8 datasets. For example, on breast-cancer, the RTE of the robust trees of Chen et al. [9] is 13.1%, while the RTE of our robust model is 6.6% and we achieve the same test error of 0.7%. We note that TE and RTE of our robust trees are in many cases better than for our robust stumps. This suggests that there is a benefit of using more expressive weak learners in boosting to get more robust and accurate models. Adversarial training performs worse than our provable defense not only in terms of URTE, but even in terms of LRTE. This is different from the neural network literature [45, 25], where adversarial training usually provides better LRTE and significantly better test error than methods providing provable robustness guarantees. However, our upper bound on the robust loss is *tight* and *tractable* and thus adversarial training should not be used as it provides only a lower bound and minimization of an upper bound makes more sense than minimization of a lower bound. We provide a more detailed comparison to Chen et al. [9] in Appendix G.2 including multi-class datasets (MNIST, FMNIST). We also show there that our proposed method to calculate the certified robust error (URTE) is orders of magnitudes faster than the MIP formulation.

**Comparison to provable defenses for neural networks** We note that our methods are primarily suitable for tabular data, but in the literature on robustness of neural networks there are no established tabular datasets to compare to. Thus, we compare our robust boosted trees to the convolutional networks of [73, 16, 75, 25, 13] on MNIST, FMNIST, and CIFAR-10. We do not compare to randomized smoothing since it is competitive only for small $l_\infty$-balls [57]. Since the considered datasets are multi-class, we extend our training of robust boosted trees from the binary classification case to multi-class using the one-vs-all approach described in Appendix E. We also use data augmentation by shifting the images by one pixel horizontally and vertically. We fit our robust trees with depth of up to 30 for MNIST and FMNIST, and with depth of up to 4 for CIFAR-10. Note that we restrict the minimum number of examples in a leaf to 100. Thus a tree of depth 30 makes only a small fraction of the possible $2^{30}$ splits. We provide a comparison in Table 3. In terms of provable robustness (URTE), our method is competitive to many provable defenses for CNNs. In particular, we outperform the LP-relaxation approach of [73] on all three datasets both in terms of test error and upper bounds. We also systematically outpeform the recent approach of [75] aiming at enhancing verifiability of CNNs – we have a better URTE with the same or better test error. Only the recent work of [25] is able to outperform our approach. Also, the CIFAR-10 model of [16] shows better URTE than our approach, but worse test error. We would like to emphasize that even on CIFAR-10 (with a relatively large $\epsilon = 8/255$) our models are not too far away from the state-of-the-art. In addition our robust boosted tree models require less computations at inference time.

**Robustness vs accuracy tradeoff** There is a lot of empirical evidence that robust training methods for neural networks exhibit a trade-off between robustness and accuracy [73, 25, 64]. We can confirm that the trade-off can also be observed for boosted trees: we consistently lose accuracy once we increase $\epsilon$. The only *slight* gain in accuracy that we observe is on FMNIST shoes dataset. More details and plots of robustness versus accuracy can be found in Appendix G.4.

**Table 3:** Comparison of our robust boosted trees to the state-of-the-art provable defenses for convolutional neural networks reported in the literature. Our models are competitive to them in terms of upper bounds on robust test error (URTE). By * we denote results taken from [25] where they could achieve significantly better TE and URTE with the code of [73].

| Dataset | $l_\infty \epsilon$ | Approach | TE | LRTE | URTE |
|---|---|---|---|---|---|
| MNIST | 0.3 | Wong et al. [73]* | 13.52% | 26.16% | 26.92% |
| | | Xiao et al. [75] | 2.67% | 7.95% | 19.32% |
| | | **Our robust trees, depth 30** | 2.68% | 12.46% | 12.46% |
| | | Gowal et al. [25] | 1.66% | 6.12% | **8.05%** |
| FMNIST | 0.1 | Wong and Kolter [72] | 21.73% | 31.63% | 34.53% |
| | | Croce et al. [13] | 14.50% | 26.60% | 30.70% |
| | | **Our robust trees, depth 30** | 14.15% | 23.17% | **23.17%** |
| CIFAR-10 | 8/255 | Xiao et al. [75] | 59.55% | 73.22% | 79.73% |
| | | Wong et al. [73] | 71.33% | – | 78.22% |
| | | **Our robust trees, depth 4** | 58.46% | 74.69% | 74.69% |
| | | Dvijotham et al. [16] | 59.38% | 67.68% | 70.79% |
| | | Gowal et al. [25] | 50.51% | 65.23% | **67.96%** |

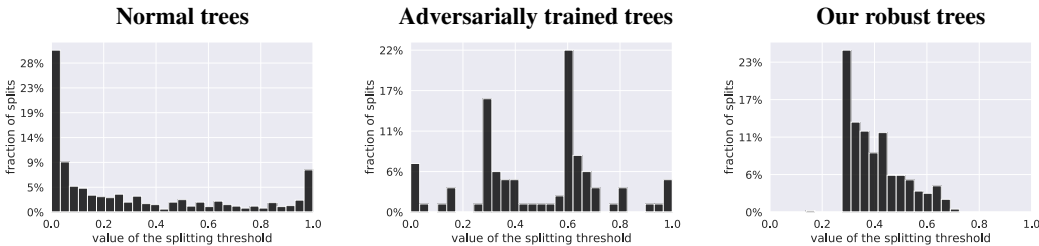

**Figure 3:** The distribution of the splitting thresholds for boosted trees models trained on MNIST 2-6. We can observe that our robust model almost always selects splits in the range between 0.3 and 0.7, which is reasonable given $l_\infty$-perturbations within $\epsilon = 0.3$. At the same time, the normal and adversarially trained models split close to 0 or 1, which suggests that their decisions might be easily flipped by the adversary.

**Interpretability** For boosted stumps or trees, unlike for neural networks, we can *directly* inspect the model and the classification rules it has learned. In particular, in Figure 3, we plot the distibution of the splitting thresholds $b$ for the three boosted trees models on MNIST 2-6 reported in Table 2. We can observe that our robust model almost always selects splits in the range between 0.3 and 0.7, which is reasonable given that more than 80% pixels of MNIST are either 0 or 1, and the considered $l_\infty$-perturbations are within $\epsilon = 0.3$. At the same time, the normal and adversarially trained models split arbitrarily close to 0 or 1, which suggests that their decisions might be easily flipped if the adversary is allowed to change them within this $\epsilon$. To emphasize the importance of interpretability and transparent decision making, we provide feature importance plots and more histograms of the splitting thresholds in Appendix G.5 and G.6.

## 6 Conclusions and Outlook

Our results show that the proposed methods achieve state-of-the-art provable robustness among boosted stumps and trees, and are also competitive to provably robust CNNs. This can be seen as a strong indicator that particularly for large $l_\infty$-balls, current provably robust CNNs are so over-regularized that their performance is comparable to simple decision tree ensembles that make decisions based on individual pixel values. Thus it remains an open research question whether it is possible to establish tight and tractable upper bounds on the robust loss for neural networks. On the contrary, as shown in this paper, for boosted decision trees there exist simple and tight upper bounds which can be efficiently optimized. Moreover, for boosted decision stumps one can compute and optimize the exact robust loss. We thus think that if provable robustness is the goal then our robust decision stumps and trees are a promising alternative as they not only come with tight robustness guarantees but also are much easier to interpret.

## Acknowledgements

We thank the anonymous reviewers for very helpful and thoughtful comments. We acknowledge the support from the German Federal Ministry of Education and Research (BMBF) through the Tübingen AI Center (FKZ: 01IS18039A). This work was also supported by the DFG Cluster of Excellence "Machine Learning – New Perspectives for Science", EXC 2064/1, project number 390727645, and by DFG grant 389792660 as part of TRR 248.

## Footnotes

[1]The order is very important as a min-max problem is not the same as a max-min problem.

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
