[Supplementary Material]

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

Now we observe that $\min_{\tilde{x} \in C} \langle r(\tilde{x}), u \rangle$ is a concave function as a pointwise minimum of a set of concave (linear) functions (see [4] regarding this property). The convexity of $\tilde{L}$ follows from the fact that it is a composition of a convex, nonincreasing function $L$ and a concave function $c + \min_{\tilde{x} \in C} \langle r(\tilde{x}), u \rangle$. $\qquad\qquad\square$

# B  Detailed algorithms

## B.1  The efficient exact certification for boosted stumps

---

**Algorithm 1:** The efficient exact certification for boosted stumps

---

**Input:** ensemble of stumps $\{f_i\}_{i=1}^T$, point $x \in \mathbb{R}^d$, label $y \in \{-1, 1\}$, radius of $l_\infty$-ball $\epsilon$
**Output:** $is\_robust \in \{0, 1\}$

1  $G \leftarrow 0$ /* initialize the variable that will be the solution of (3)     */
2  **for** $k \leftarrow 1$ **to** $d$ **do**
3  $\quad$ $\mathcal{F} = \{f \in \{f_i\}_{i=1}^T \mid c_s = k\}$ /* all stumps that split coord. $k$     */
4  $\quad$ $\delta_k^* = \texttt{CalculateMinimizer}G_k(\mathcal{F}, x, y, \epsilon)$
5  $\quad$ $G \leftarrow G + \mathcal{F}(x_k + \delta_k^*)$
6  **end**
7  $is\_robust = \mathbb{1}_{G \geq 0}$
8
9  **Function** $\texttt{CalculateMinimizer}G_k(\mathcal{F}, x, y, \epsilon)$
10  $\quad$ $\mathcal{F} \leftarrow$ merge the stumps in $\mathcal{F}$ with the same splitting thresholds
11  $\quad$ $B \leftarrow \{x_k - \epsilon\}$, $W \leftarrow \{0\}$
12  $\quad$ **for** $s \leftarrow 1$ **to** $|\mathcal{F}|$ **do**
$\quad\quad$ /* add all thresholds and weights $w_r$ in $[x_k - \epsilon, x_k + \epsilon]$     */
13  $\quad\quad$ $b_s \leftarrow \mathcal{F}_s.b$,  $w_r^{(s)} \leftarrow \mathcal{F}_s.w_r$
14  $\quad\quad$ **if** $x_k - \epsilon < b_s \leq x_k + \epsilon$ **then**
15  $\quad\quad\quad$ $B \leftarrow B \cup \{b_s\}, W \leftarrow W \cup \{w_r^{(s)}\}$
16  $\quad$ **end**
$\quad$ /* sorting thresholds in $B$ leads to $O(T \log T)$ complexity     */
17  $\quad$ $\pi = \operatorname{argsort}(B)$
18  $\quad$ $v^* \leftarrow 0$ /* initialize the minimum cumulative difference     */
19  $\quad$ $\delta_k^* \leftarrow -\epsilon$ /* initialize the optimal perturbation for coord. $k$     */
20  $\quad$ **for** $i \leftarrow 1$ **to** $|\pi|$ **do**
21  $\quad\quad$ $v \leftarrow v + y\,W_{\pi_i}$
22  $\quad\quad$ **if** $v < v^*$ **then**
23  $\quad\quad\quad$ $v^* \leftarrow v, \delta_k^* \leftarrow B_{\pi_i} - x_k$
24  $\quad$ **end**
25  $\quad$ **return** $\delta_k^*$
26  **end**

---

## B.2  Exact adversarial examples for boosted stumps

Using the result from Section 3.1, we can directly obtain provably minimal adversarial examples. By noting that the function $H(\epsilon) := \min_{\|\delta\|_\infty \leq \epsilon} yF(x + \delta)$ is piece-wise constant with up to $T + 1$ constant

regions, it suffices to solve this minimization problem for every $\epsilon \in \{0\} \cup \{|b_t - x_{c_t}| + \nu \operatorname{sign}(b_t - x_{c_t}) \mid t = 1, \ldots, T\}$ (where $\nu$ is as small as precision allows) sorted in ascending order and stop when $\epsilon$ is large enough to change the original class. In order to get the final perturbation vector $\delta$, we have to save the indices $\delta_j^*$ that minimize $yF(x + \delta)$ for every splitting coordinate $j$ which are used in the ensemble. The complexity of this procedure is $O(T^2 \log T)$ since in the worst case we have to solve (4) $T$ times. For details of the procedure we refer to Algorithm 2. We provide visualizations of these exact adversarial examples in Figure 11.

---

**Algorithm 2:** Finding exact adversarial examples for boosted stumps

---

**Input:** ensemble of stumps $\{f_i\}_{i=1}^T$, point $x \in \mathbb{R}^d$, label $y \in \{-1, 1\}$
**Output:** exact adversarial perturbation $\delta \in \mathbb{R}^d$

1  $\mathcal{E} \leftarrow \{0\} \cup \{|b_t - x_{c_t}| + \nu \operatorname{sign}(b_t - x_{c_t}) \mid t = 1, \ldots, T\}$
2  $\mathcal{E} \leftarrow \operatorname{sort}(\mathcal{E})$
3  **for** $i \leftarrow 1$ **to** $|\mathcal{E}|$ **do**
4      $\epsilon \leftarrow \mathcal{E}_i$
5      $G_\epsilon \leftarrow 0$
6      $\delta \leftarrow \mathbf{0}$ /* initialize the adversarial perturbation                   */
7      **for** $k \leftarrow 1$ **to** $d$ **do**
8          $\mathcal{F} = \{f \in \{f_i\}_{i=1}^T \mid c_s = k\}$ /* all stumps that split coord. $k$   */
9          $\delta_k^* \leftarrow \texttt{CalculateMinimizer}G_k(\mathcal{F}, x, y, \epsilon)$ /* from Algorithm 1   */
10         $G_\epsilon \leftarrow G_\epsilon + \mathcal{F}(x_k + \delta_k^*)$
11     **end**
12     **if** $G_\epsilon < 0$ **then**
13         **break**
14 **end**

---

### B.3 Coordinate descent for the exponential loss

**Boosted decision stumps:** If we denote $\mathbb{1}_i$ to be equal to 1 if the first condition of (5) is true, and 0 otherwise, and also define

$$\gamma_i = \exp\Big(-\sum_{k \neq j} G_k(x_i, y_i) - \sum_{s \in S_j} y_i w_l^{(s)} - h_r(x_{ij}, y_i)\mathbb{1}_i - h_l(x_{ij}, y_i)(1 - \mathbb{1}_i)\Big),$$

then the total exponential loss can be written as $L(w_l, w_r) = \sum_{i=1}^n \gamma_i \exp\big(-y_i w_l - y_i w_r \mathbb{1}_i\big)$. We further denote $\mathbb{1}_{y_i=y} = \begin{cases} 1 & \text{if } y_i = y \\ 0 & \text{if } y_i \neq y \end{cases}$ and

$$\Sigma_{1,1} = \sum_{i=1}^n \mathbb{1}_i \mathbb{1}_{y_i=1} \gamma_i \qquad \Sigma_{1,-1} = \sum_{i=1}^n \mathbb{1}_i \mathbb{1}_{y_i=-1} \gamma_i \qquad (12)$$

$$\Sigma_{0,1} = \sum_{i=1}^n (1 - \mathbb{1}_i)\mathbb{1}_{y_i=1} \gamma_i \quad \Sigma_{0,-1} = \sum_{i=1}^n (1 - \mathbb{1}_i)\mathbb{1}_{y_i=-1} \gamma_i$$

Then the coordinate descent update for $w_l$ can be derived by setting $\frac{\partial L}{\partial w_l}$ to zero and solving for $w_l$ which yields:

$$w_l = \frac{1}{2}\ln\big(\exp(-w_r)\Sigma_{1,1} + \Sigma_{0,1}\big) - \frac{1}{2}\ln\big(\exp(w_r)\Sigma_{1,-1} + \Sigma_{0,-1}\big)$$

Thus, the overall complexity for a particular coordinate $j$ and fixed threshold $b$ is $O(n)$ times the number of iterations of coordinate descent which is logarithmic in the desired precision (cost for bisection).

**Boosted decision trees:** By using the notation from (12), where now $\mathbb{1}_i := \mathbb{1}(x_i, y_i; w_r)$, the minimizers of $w_r$ and $w_l$ are given by setting $\frac{\partial L}{\partial w_r}$ and $\frac{\partial L}{\partial w_l}$ to zero:

$$w_r = \frac{1}{2} \ln(\Sigma_{1,1}) - \frac{1}{2} \ln(\Sigma_{1,-1}) - w_l$$

$$w_l = \frac{1}{2} \ln\left(\exp(-w_r)\Sigma_{1,1} + \Sigma_{0,1}\right) - \frac{1}{2} \ln\left(\exp(w_r)\Sigma_{1,-1} + \Sigma_{0,-1}\right)$$

We iterate these updates of $w_r$ and $w_l$ until convergence. Note that coordinate descent does not create a significant overhead to the overall algorithm, since we perform only operations on scalars $\Sigma_{1,1}$, $\Sigma_{1,-1}, \Sigma_{0,1}, \Sigma_{0,-1}$ which do not have to be recomputed over the iterations of the coordinate descent.

### B.4   Tree-wise certification of boosted decision trees

---

**Algorithm 3:** Tree-wise certification of boosted decision trees

---

**Input:** tree ensemble $\{f_t\}_{t=1}^T$, point $x \in \mathbb{R}^d$, label $y \in \{-1, 1\}$
**Output:** $is\_provably\_robust \in \{0, 1\}$

1 $\tilde{G} = 0$
2 **for** $t \leftarrow 1$ **to** $T$ **do**
3 $\quad$ $\tilde{G} = \tilde{G} + \texttt{ExactTreeCertification}(f_t, x, y)$
4 **end**
5 $is\_provably\_robust = \mathbb{1}_{\tilde{G} \geq 0}$

6 **Function** $\texttt{ExactTreeCertification}(f, x, y)$
$\quad$ /* start from a set that contains only the root node $f$ $\quad\quad$ */
7 $\quad$ $nodes\_to\_check = \{f\}$
8 $\quad$ $v^* = \infty$
9 $\quad$ **while** $nodes \neq \{\}$ **do**
$\quad\quad$ /* retrieve a node and delete it from the set $\quad\quad\quad\quad$ */
10 $\quad\quad$ $node = nodes.pop()$
$\quad\quad$ /* get the splitting coordinate of the current node $\quad\quad$ */
11 $\quad\quad$ $j = node.split\_coordinate$
12 $\quad\quad$ **if** $x_j \leq b + \epsilon$ **then**
13 $\quad\quad\quad$ **if** $node.left$ $is$ $empty$ **then**
14 $\quad\quad\quad\quad$ $v^* \leftarrow \min(v^*, y \cdot node.w_l)$
15 $\quad\quad\quad$ **else**
16 $\quad\quad\quad\quad$ $nodes\_to\_check \leftarrow nodes\_to\_check \cup \{node.left\}$
17 $\quad\quad$ **if** $x_j \geq b - \epsilon$ **then**
18 $\quad\quad\quad$ **if** $node.right$ $is$ $empty$ **then**
19 $\quad\quad\quad\quad$ $v^* \leftarrow \min(v^*, y \cdot node.w_l + y \cdot node.w_r)$
20 $\quad\quad\quad$ **else**
21 $\quad\quad\quad\quad$ $nodes\_to\_check \leftarrow nodes\_to\_check \cup \{node.right\}$
22 $\quad$ **end**
23 $\quad$ **return** $v^*$
24 **end**

---

## C Monotonic descent of the upper bound on the robust loss with the shrinkage parameter

As introduced in [23], the shrinkage parameter is applied during training as follows. Let $f$ be a new weak learner, then instead of adding it directly to the ensemble $F := F + f$, one rather adds $F := F + \alpha f$ where $\alpha \in (0, 1]$. In order to show that this scheme also always leads to monotonic descent of the upper bound on the robust loss, we apply Lemma 1 to the case where $f$ is a decision tree with $l$ leaves, i.e. $f(x) = u_{q(x)}$. Note that:

$$\tilde{L}(u) = \max_{\tilde{x} \in B_\infty(x,\epsilon)} L(\tilde{G}(x,y) + y u_{q(\tilde{x})}) = \max_{\tilde{x} \in C} L(c + \langle r(\tilde{x}), u \rangle),$$

where $c = \tilde{G}(x, y)$ is the contribution of the previous weak learners (see Equation (8)), $r(x) \in \{-1, 0, 1\}^l$ represents mutually exclusive boolean conditions of the tree $f$ multiplied by the label $y$, i.e. $r(x)_{q(x)} = y$ and $r(x)_i = 0$ for every $i \neq q(x)$. Thus, the robust loss $\tilde{L}(u)$ is convex in the leaf weights $u$.

Note that $\tilde{L}(\mathbf{0})$ corresponds to the loss value when all weights of the new weak learner $f$ are set to zero, thus it is simply the loss of the previous ensemble. Since $\tilde{L}(u)$ is convex in its leaf weights $u \in \mathbb{R}^l$, the following property holds for every $\alpha \in (0, 1]$ due to convexity of $\tilde{L}$:

$$\tilde{L}(u) < \tilde{L}(\mathbf{0}) \implies \tilde{L}(\alpha u) < \tilde{L}(\mathbf{0})$$

To see this, from the definition of convexity we have:

$$\tilde{L}(\alpha u + (1 - \alpha)\mathbf{0}) \leq \alpha \tilde{L}(u) + (1 - \alpha)\tilde{L}(\mathbf{0})$$
$$\tilde{L}(\alpha u) \leq \alpha(\tilde{L}(u) - \tilde{L}(\mathbf{0})) + \tilde{L}(\mathbf{0})$$
$$\tilde{L}(\alpha u) < \tilde{L}(\mathbf{0})$$

Moreover, since the sum of losses over training points is also convex in $u$, the same reasoning applies to the sum of upper bounds on the robust losses taken over the training set. Thus, we conclude that the usage of the shrinkage parameter $\alpha$ still preserves the monotonic descent in the robust objective, therefore its usage is justified within our robust optimization framework.

## D The cube attack

In the main part we described how to efficiently compute *upper bounds* on the robust test error. Now we would like to also have an efficient $l_\infty$ adversarial attack on boosted trees that would allow us to perform adversarial training. Moreover, it is also interesting to visualize adversarial examples to get a better understanding how the model makes its decisions. Concretely, the goal is to find $\delta \in \mathbb{R}^d$ that approximately minimizes the following optimization problem:

$$\min_{\|\delta\|_\infty \leq \epsilon} yF(x + \delta). \tag{13}$$

We note that while there is a vast literature of black-box adversarial attacks evaluated on neural networks [6, 30, 11, 26], query-efficiency of black-box $l_\infty$ attacks on boosted trees is less studied [11, 9]. In this paper we do not aim to fully explore this direction since our goal is primarily *provable* robustness, i.e. how to derive and optimize *upper bounds* on the robust error. Therefore, we just introduce a simple black-box attack that empirically works well for boosted trees and is efficient enough to be applied in adversarial training. We call it *the cube attack* which is based on (1+1) evolutionary algorithm [14]. The main idea of the proposed attack is that on every iteration we try to change some random subset of coordinates and accept the change only if it decreases the functional margin $yF(\hat{x})$ for the perturbed point $\hat{x}$. On every iteration of the attack, a potential change for every coordinate is sampled randomly from $\delta_i \in \{-2\epsilon, 0, 2\epsilon\}$, and after adding such $\delta \in \mathbb{R}^d$ to the perturbed point $\hat{x}_{new} := \hat{x} + \delta$ we do a projection s.t. $\|\hat{x}_{new}\|_\infty \leq \epsilon$ (and for images also $\hat{x}_{new} \in [0, 1]^d$) is satisfied. After this we keep $\hat{x}_{new}$ if $yF(\hat{x}_{new}) < yF(\hat{x})$, otherwise we keep the old value $\hat{x}$. The full procedure is specified in Algorithm 4.

Note that the obtained adversarial example is always situated at a corner of the feasible set (which is a cube or the intersection of two cubes for image data, and hence the name of the attack). A similar

idea of considering only corners of the feasible set was also used in [46] where they could design a successful adversarial attack for neural networks. The only obvious disadvantage of this attack is that it is restricted only to the corners of the $l_\infty$-ball. However, since the considered $l_\infty$-balls are small, it is unlikely to have a decision region which crosses only the interior of the ball, but none of its corners. The tight lower bounds on the robust test error that we show in our experiments suggest that this is indeed true in practice. Moreover, for many models the lower and upper bounds on the robust test error are *exactly equal* which suggests that with the proposed method we can avoid using expensive combinatorial MIP solvers for large-scale classification tasks while still being able to accurately estimate the robustness of the models.

---

**Algorithm 4:** The cube attack

---

**Input:** classifier $F$, point $x \in \mathbb{R}^d$, label $y \in \{-1, 1\}$, number of iterations $N$, probability $p$ to
change a coordinate (default value: $p = 0.5$)
**Output:** approximate minimizer $\delta \in \mathbb{R}^d$ of (13)

1   $\hat{x} \leftarrow x$ /* initialize the adversarial example                                  */
2   $v^* \leftarrow yF(x)$ /* initialize the minimum functional margin                */
3   **for** $i \leftarrow 1$ **to** $N$ **do**
4      $\delta_i \sim$ Categorical $([-2\epsilon,\ 0,\ 2\epsilon]$ with probabilities $[p/2,\ 1-p,\ p/2])\ \ \forall i \in 1, \ldots, d$
5      $\hat{x}_{new} \leftarrow$ Projection of $\hat{x} + \delta$ onto $B_\infty(x, \epsilon)$ (for images also onto $[0, 1]^d$)
6      $v_{new} \leftarrow yF(\hat{x}_{new})$
       /* if the objective is improved, keep the new point $\hat{x}_{new}$        */
7      **if** $v_{new} < v^*$ **then**
8          $\hat{x} \leftarrow \hat{x}_{new}$
9          $v^* \leftarrow v_{new}$
10   **end**
11   $\delta \leftarrow \hat{x} - x$

---

# E   Extension of the method to multi-class setting

## E.1   Certification for multi-class setting

We assume that for a multi-class classifier $F : \mathbb{R}^d \to \mathbb{R}^K$, a point $x \in \mathbb{R}^d$ is classified using $y = \arg\max_{c=1,\ldots,K} F_c(x)$. Now if $y \in \{1, \ldots, K\}$ is the correct class, then $x$ is correctly classified if and only if

$$\min_{c \neq y} [F_y(x) - F_c(x)] > 0.$$

Then the multi-class variant of the certification procedure 3 has the following form:

$$G_{mult}(x, y) = \min_{c \neq y} \min_{\|\delta\|_p \leq \epsilon} [F_y(x + \delta) - F_c(x + \delta)] \tag{14}$$

$$= \min_{c \neq y} \min_{\|\delta\|_p \leq \epsilon} \left[ \sum_{t=1}^{T} f_{yt}(x + \delta) - \sum_{t=1}^{T} f_{ct}(x + \delta) \right]$$

And then analogously to the binary classification case, it is not possible to change the class via a perturbation within the $l_p$-ball of radius $\epsilon$ if and only if $G_{mult}(x, y) > 0$.

The crucial observation now is that in the objective of (14), we have just an ensemble of $2T$ trees ($T$ trees for each class), which we already showed how to solve exactly for stumps using Algorithm 1, and how to lower bound for trees using Algorithm 3 in order to get a robustness guarantee. Thus, robustness certification for the multi-class setting can be done directly by reusing the same routines $K - 1$ times, i.e. for every $c \in \{1, \ldots, K\} \setminus y$, and then taking the minimum over the $K - 1$ values and comparing it to zero.

## E.2   Robust training for multi-class setting

Now we discuss how to properly integrate the multi-class guarantee into training via calculating an upper bound on the robust loss.

One of the first popular multi-class versions of AdaBoost is AdaBoost.MH suggested in [58] which is essentially one-vs-all classifier if labels are mutually exclusive. Although, [23] argue that the one-vs-all scheme is suboptimal, their results show that the one-vs-all approach performs similarly to the joint cross-entropy loss, see also [37, 38] for a more recent comparison. [55] compared a wide range of multi-class methods and concluded that with proper tuning of the hyperparameters of the classifiers, one-vs-all approach does not show worse results than other more involved methods. Our experiments again confirm this observation where we found that our non-robust one-vs-all models perform similarly to the models trained with XGBoost library. Thus, we describe below how we perform provably robust training for the one-vs-all scheme.

Assuming that labels for class $c$ and training point $x_i$ are $y_{ci} \in \{-1, 1\}$, by the *one-vs-all scheme* we mean the following optimization problem:

$$\min_{F_1,\ldots,F_K} \sum_{i=1}^{n} \sum_{c=1}^{K} L(y_{ci} F_c(x_i)) = \sum_{c=1}^{K} \min_{F_c} \sum_{i=1}^{n} L(y_{ci} F_c(x_i))$$

The crucial observation is that the objective is separable over the individual classifiers $F_1, \ldots, F_K$, and thus the $K$ classifiers can be trained completely independently. A clear advantage of such a scheme is that it can be trivially parallelized. However, it is not separable anymore if we consider the *robust one-vs-all scheme* since the same adversarial perturbation $\delta$ is shared across $K$ losses. But we still can upper bound the sum of robust losses $L(y_{ci} F_c(x_i + \delta))$ element-wise:

$$\min_{F_1,\ldots,F_K} \sum_{i=1}^{n} \max_{\|\delta\|_p \leq \epsilon} \sum_{c=1}^{K} L(y_{ci} F_c(x_i + \delta)) \leq \min_{F_1,\ldots,F_K} \sum_{i=1}^{n} \sum_{c=1}^{K} \max_{\|\delta\|_p \leq \epsilon} L(y_{ci} F_c(x_i + \delta))$$

and then train $K$ one-vs-all classifiers independently. Note that from the implementation point of view, for boosted trees with the exponential loss, the only difference compared to the binary classification scheme that we described earlier is just different per-example weights $\gamma$. Thus, this scheme can be easily implemented by reusing the same procedures described earlier. This scheme already works quite well for robust boosted trees. However, we note that it is not clear how to perform *exact* robust optimization for boosted stumps for the original robust one-vs-all objective.

# F  Experimental details

**Datasets:**  All datasets used in the experiments are listed in Table 4.

**Table 4:** Information about the datasets used in the experiments.

| Dataset | # classes | # features | # train | # test | Reference |
|---|---|---|---|---|---|
| breast-cancer | 2 | 10 | 546 | 137 | [15] |
| diabetes | 2 | 8 | 614 | 154 | [60] |
| cod-rna | 2 | 8 | 59535 | 271617 | [65] |
| MNIST 1-5 | 2 | 784 | 12163 | 2027 | [40] |
| MNIST 2-6 | 2 | 784 | 11876 | 1990 | [40] |
| FMNIST shoes | 2 | 784 | 12000 | 2000 | [74] |
| GTS 100-rw | 2 | 3072 | 4200 | 1380 | [61] |
| GTS 30-70 | 2 | 3072 | 2940 | 930 | [61] |
| MNIST | 10 | 784 | 60000 | 10000 | [40] |
| FMNIST | 10 | 784 | 60000 | 10000 | [74] |
| CIFAR-10 | 10 | 3072 | 50000 | 10000 | [35] |

**Hyperparameters;**  For the breast-cancer dataset, we select the radius $\epsilon$ of the $l_\infty$-perturbations based on the choice of [9]. However, for diabetes and cod-rna datasets we reduce them compared to [9] in a way that allows robust classifiers to still achieve a test error comparable to normal models. For image datasets (MNIST, FMNIST, GTS, CIFAR-10), we follow the established $l_\infty$ $\epsilon$'s from the neural networks literature [72, 25].

We tune the hyperparameter $w_{max}$ on the validation sets of several datasets and the best value came out to be close to 1, which we use for all experiments. For binary classification, we use the shrinkage

parameter of 0.2 for diabetes, cod-rna, MNIST 1-5, MNIST 2-6, and FMNIST shoes, and 0.01 for the rest of the datasets. We use at most 300 iterations for stumps, 300 iterations for trees of depth 2, 150 iterations for depth 4, and 75 iterations for trees of depth 8. For trees, we perform splits only when there are more than 10 examples at a leaf for binary classification datasets, and if more than 200 examples for multi-class datasets.

**Restricting the maximum weight:**   In the process of fitting a decision stump (also as an intermediate step for building a tree), we have to take care of cases when all points at some side of the threshold $b$ have the same label. This leads to $w_l$ or $w_r$ that attain their optimal values at $\pm\infty$ depending on the labels. In order to resolve this, in our implementation we set the maximum weight $w_{max}$, and we project all obtained leaf values $w_l$ and $w_l + w_r$ onto the range $[-w_{max}, w_{max}]$. We found empiricially that constraining the maximum values of tree leafs in this way leads to a noticeable beneficial regularization effect which is similar in spirit to the usage of the shrinkage parameter introduced in [22].

## G   Additional experiments

### G.1   Adversarial training for boosted stumps

We show the results of adversarial training for boosted stumps in in Table 5, where adversarial examples were generated using the cube attack with 10 iterations and $p = 0.5$. We observed that we could achieve non-trivial robustness (RTE) with adversarially trained models only when we used a small shrinkage parameter. Thus, we set it to 0.1 for all boosted stump models.

The results show that similarly to boosted trees, both robust training of Chen et al. [9] and our proposed methods outperform adversarial training by a large margin. This shows that either one has to find a better way to perform adversarial training for boosted stumps and trees, or that it may not be a suitable technique for classifiers which are built in a stagewise fashion.

### G.2   Comparison to the robust training of Chen et al. [9]

We compare our provably robust boosted stumps and trees to Chen et al. [9] in the same setting as ours: we fit boosted stumps and boosted trees of depth 4 with 80% of the training data and use the rest as the validation set for model selection. For the models of Chen et al. [9] we use exact RTE via MIP of [32] for model selection both for stumps and trees, whereas for our models we use exact RTE for stumps, and our fast URTE for trees. For [9] we use a coarser grid for selecting the number of iterations since RTE, in particular for trees, is more expensive to evaluate. We use up to 300 iterations and shrinkage parameter of 1 for boosted stumps both for us and [9]. For boosted trees of [9] we use the number of iterations and the shrinkage parameters for every dataset separately as specified in the code of [9], and for our models as described in the previous section.

**Boosted stumps:**   We present the results in Table 6. We can see that our robust trees lead to better RTE on 7 out of 8 datasets while having comparable test error. Moreover, our efficient way of

**Table 5:** The results of adversarially trained boosted stumps, where adversarial examples were generated using the cube attack. The results for other training methods are presented in Table 1. We conclude that our proposed robust stumps outperform adversarial training by a large margin.

| Dataset | $l_\infty \; \epsilon$ | Adversarially trained stumps | | |
|---|---|---|---|---|
| | | TE | RTE | URTE |
| breast-cancer | 0.3 | 0.7 | 15.3 | 15.3 |
| diabetes | 0.05 | 27.3 | 33.1 | 33.1 |
| cod-rna | 0.025 | 22.8 | 26.1 | 26.1 |
| MNIST 1-5 | 0.3 | 3.2 | 8.3 | 9.1 |
| MNIST 2-6 | 0.3 | 9.7 | 22.5 | 24.6 |
| FMNIST shoes | 0.1 | 8.3 | 16.3 | 17.0 |
| GTS 100-rw | 8/255 | 2.2 | 7.7 | 7.9 |
| GTS 30-70 | 8/255 | 19.1 | 28.8 | 31.0 |

**Table 6:** Comparison of our boosted stumps to Chen et al. [9]. The model selection of the number of iterations *#iter* was done based on RTE. *Time MIP* and *Time ours* correspond to the time needed to calculate RTE of our models using a general-purpose MIP solver and our fast exact certification procedure described in Section 4.1. All numbers are obtained using full test sets.

| Dataset | $l_\infty$ $\epsilon$ | Stumps of Chen et al. [9] | | | Our robust stumps (exact robust loss) | | | | | |
|---|---|---|---|---|---|---|---|---|---|---|
| | | TE | RTE | #iter | TE | RTE | #iter | Time MIP | Time ours | Speedup |
| breast-cancer | 0.3 | 8.8 | 16.8 | 1 | 5.1 | **10.9** | 2 | 0.1s | **0.2ms** | **529x** |
| diabetes | 0.05 | 23.4 | **30.5** | 3 | 27.3 | 31.8 | 1 | 0.1s | **0.3ms** | **393x** |
| cod-rna | 0.025 | 11.6 | 23.2 | 4 | 11.2 | **22.6** | 16 | 6.5m | **69ms** | **5655x** |
| MNIST 1-5 | 0.3 | 0.9 | 5.2 | 40 | 0.7 | **3.6** | 274 | 37.7s | **0.14s** | **267x** |
| MNIST 2-6 | 0.3 | 2.8 | 13.9 | 40 | 3.0 | **9.2** | 83 | 14.5s | **48ms** | **302x** |
| FMNIST shoes | 0.1 | 7.1 | 22.2 | 10 | 5.7 | **10.8** | 174 | 23.9s | **91ms** | **260x** |
| GTS 100-rw | 8/255 | 2.0 | 11.8 | 40 | 2.0 | **6.7** | 109 | 8.1s | **0.10s** | **80x** |
| GTS 30-70 | 8/255 | 12.7 | 28.2 | 40 | 12.9 | **27.6** | 227 | 20.9s | **0.47s** | **45x** |

**Table 7:** Comparison of our boosted trees to Chen et al. [9]. The model selection of the number of iterations *#iter* was done based on RTE for the models of [9] and URTE for our models. *Time MIP* and *Time ours* correspond to the time needed to calculate RTE of our models using a MIP solver and URTE as described in Section 4.2. All numbers are obtained using full test sets.

| Dataset | $l_\infty$ $\epsilon$ | Trees of Chen et al. [9] | | | Our robust trees (robust loss bound) | | | | | | |
|---|---|---|---|---|---|---|---|---|---|---|---|
| | | TE | RTE | #iter | TE | RTE | URTE | #iter | Time MIP | Time ours | Speedup |
| breast-cancer | 0.3 | 0.7 | 13.1 | 8 | 0.7 | **6.6** | 6.6 | 46 | 5.8s | **12ms** | **502x** |
| diabetes | 0.05 | 22.1 | 40.3 | 5 | 27.3 | **35.7** | 35.7 | 9 | 1.1s | **3ms** | **343x** |
| cod-rna | 0.025 | 10.2 | 24.2 | 20 | 6.9 | **21.3** | 21.4 | 36 | 31.9m | **3.5s** | **550x** |
| MNIST 1-5 | 0.3 | 0.3 | 2.9 | 1000 | 0.2 | **1.3** | 1.4 | 126 | 3.7m | **0.14s** | **1581x** |
| MNIST 2-6 | 0.3 | 0.5 | 6.9 | 1000 | 0.7 | **3.8** | 4.1 | 88 | 2.6m | **0.10s** | **1500x** |
| FMNIST shoes | 0.1 | 3.1 | 13.2 | 20 | 3.6 | **8.0** | 8.1 | 128 | 3.6m | **0.14s** | **1522x** |
| GTS 100-rw | 8/255 | 1.5 | 9.7 | 20 | 2.6 | **4.7** | 4.7 | 105 | 1.4m | **57ms** | **1417x** |
| GTS 30-70 | 8/255 | 11.5 | 28.8 | 20 | 13.8 | **20.9** | 21.4 | 148 | 2.4m | **0.10s** | **1463x** |
| MNIST | 0.3 | 2.0 | 31.2 | 200 | 2.7 | **12.5** | 15.8 | 37 | 5.5 days | **4.6s** | **135893x** |
| FMNIST | 0.1 | 14.4 | 65.1 | 200 | 14.2 | **23.2** | 25.9 | 52 | 3.3 days | **4.2s** | **82209x** |

calculating the RTE described in Section 3.1 is orders of magnitude faster than using an off-the-shelf MIP-solver [27]. We note that the most robust models of [9] are usually obtained at the first 40 iterations, while our models need more iterations to obtain the minimum validation RTE. We attribute this to the differences in robust training and also to the fact that we use a different loss function and constrain $w_{max}$. We observe that the latter usually increases the number of iterations needed until convergence.

**Boosted trees:** First, we note that in order to make the MIP formulation of [32] more scalable for tree ensembles, we change it to the feasibility problem regarding whether there exists an $l_\infty$-perturbation that is able to change the class instead of searching for the *minimal* adversarial perturbation wrt the $l_\infty$-norm. This brings us in average two orders of magnitude speed-up for calculating RTE on the considered datasets. However, even with this speed-up, it still takes up to 5.5 days to calculate RTE for the largest models that we evaluated.

We present the comparison for boosted trees in Table 7. The main observation is that we outperform [9] on *all* considered datasets in terms of the RTE, often by a large margin. Our better RTE comes at the price of slightly worse test error on several datasets which we attribute to the empirically observed trade-off between accuracy and robustness: methods achieving better robustness tend to have worse test error. We note that our URTE are very close to RTE, and the time needed to calculate URTE is orders of magnitude faster than RTE calculated with MIP.

In Table 7 we also provide a comparison for boosted trees on multi-class datasets (MNIST and FMNIST). We trained our models using the one-vs-all approach and set the depth of individual trees to be up to 30. For [9] we take the models provided by the authors that have depth 8. We can see that our robust trees outperform their method by a large margin: 12.5% instead of 31.2% RTE on MNIST. On FMNIST, the gap is even larger: 23.2% versus 65.1% RTE while our test error is even slightly

better. We note that partially the reason for such a large gap might be in the fact that the boosted trees of [9] may also benefit from a larger depth. However, our comparison for binary classification datasets suggests that even when the settings are the same for both methods, our robust training consistently leads to more robust models than [9].

### G.3 Robust boosted trees of different depth

The results for boosted trees are given in Table 8 for trees of different depth. We show lower bounds on robust test error (LRTE) obtained via the cube attack to show that it leads to tight LRTE which are close to the exact RTE values. This justifies its usage in adversarial training. For LRTE we used the attack with 20 iterations and $p = 0.5$. We run the attack every iteration of training, and initialize every next perturbation $\delta$ with the perturbation obtained at the previous iteration. We perform $l_\infty$ adversarial training similarly to [32], i.e. every iteration we train on clean training points and adversarial examples (equal proportion), which are generated via the cube attack using 10 iterations and $p = 0.5$.

We observe that robust training for boosted trees is very efficient in improving robustness of the models for all depth values. In particular, our robust models outperform adversarially trained models, often with a large margin. For example, on MNIST 1-5, RTE of the adversarially trained model of depth 8 is $10.5\%$, while RTE of our robust model of the same depth is $1.2\%$. We observe that for our robust trees, URTE is very close to LRTE or even the same in some cases which can allow us to assess exact RTE even without using any combinatorial solvers. Finally, we note that our trees of depth 4 outperform our trees of depth 2 on all datasets in terms of RTE. However, our models of depth 8 show a better RTE than depth 4 only on several datasets including MNIST 1-5 and MNIST 2-6. For MNIST and FMNIST we observed improvements in RTE by increasing the depth up to 30. This suggests that in order to achieve the optimal RTE, one has to carefully select an appropriate depth of the trees which depends on a particular dataset.

### G.4 Robustness and accuracy

There is a lot of empirical evidence that robust training methods for neural networks exhibit a trade-off between robustness and accuracy [73, 25, 64]. Now we investigate whether the same trade-off also exists for our robust boosted trees. For this we take three datasets (diabetes, cod-rna, and FMNIST shoes) and plot the dependency of the test error on $l_\infty$ $\epsilon$ used for our robust training. The results are presented in Figure 4 for trees of depth 4 and 8. We can confirm that the trade-off can also be observed for boosted trees: we consistently lose accuracy once we increase $\epsilon$. The only *slight* gain in accuracy that we observe is on FMNIST shoes dataset.

### G.5 Feature importance

It is important to note that boosted trees that split directly on pixel values are not the most suitable models for computer vision tasks. Even though on some datasets like GTS 100-rw, they are able to achieve less than 1% test error, they lack important invariances such as invariance to translations, different view points, etc. What we would like to emphasize in this section is the advantage of boosted trees in terms of *transparent decision making*. In particular, we can clearly see which pixels are directly used for the decisions. One of the ways to assign feature importance to boosted decision trees with coordinate-aligned splits is to count the number of times a particular feature was used in some splits. Such visualization are shown in Figures 5, 6, 7. First of all, we can note that for all datasets our robust training changes the frequencies of features that are used. For example, on the breast cancer dataset, the robust model tends to use features like texture, concave points, area, radius, and compactness much less often compared to the normal and adversarially trained models. On MNIST 1-5 and MNIST 2-6 we see that the robust model relies more often at the pixels which are closer to the border. On GTS 100-rw and GTS 30-70 all the models rely mainly just on a few discriminative pixels (see Figure 13 for examples of the images). It is particularly interesting that on GTS 100-rw the models can achieve almost perfect classification error while ignoring almost the whole image. This shows that even a good performance on some test set does not yet mean that the model has truly learned important features – just shifting the GTS images by several pixel would completely ruin the performance of the presented boosted tree models. Thus we again emphasize the importance of interpretability for detecting such failure modes.

**Table 8:** Evaluation of robustness for boosted trees of different depth. We show, in percentage, test error (TE), lower bound on robust test error (LRTE) via the cube attack, robust test error (RTE) via MIP of [32], upper bound on robust test error (URTE), and the number of iterations selected using the validation set (#iter). Our robust boosted trees significantly improve RTE, more than adversarially trained boosted trees. We also observe that URTE is close to RTE for many models.

| Dataset | $l_\infty\ \epsilon$ | Normal trees (standard training) | | | | | Adversarially trained trees (with cube attack) | | | | | Our robust trees (robust loss bound) | | | | |
|---|---|---|---|---|---|---|---|---|---|---|---|---|---|---|---|---|
| | | TE | LRTE | RTE | URTE | #iter | TE | LRTE | RTE | URTE | #iter | TE | LRTE | RTE | URTE | #iter |
| **depth=2** | | | | | | | | | | | | | | | | |
| breast-cancer | 0.3 | 1.5 | 81.0 | 81.0 | 82.5 | 47 | **0.7** | 29.2 | 29.2 | 29.2 | 3 | 2.2 | **10.2** | **10.2** | **10.2** | 12 |
| diabetes | 0.05 | **22.7** | 43.5 | 44.8 | 45.5 | 20 | 25.3 | 38.3 | 38.3 | 38.3 | 3 | 28.6 | **36.4** | **36.4** | **36.4** | 20 |
| cod-rna | 0.025 | **3.9** | 35.6 | 37.0 | 39.2 | 298 | 11.5 | 22.9 | 22.9 | 22.9 | 2 | 7.2 | **21.6** | **21.6** | **21.6** | 229 |
| MNIST 1-5 | 0.3 | **0.1** | 57.5 | 88.5 | 99.0 | 192 | 1.9 | 8.6 | 8.8 | 9.1 | 7 | 0.5 | **1.8** | **1.8** | **1.8** | 140 |
| MNIST 2-6 | 0.3 | **0.7** | 95.5 | 100 | 100 | 276 | 4.7 | 17.5 | 17.5 | 17.5 | 8 | 1.2 | **4.8** | **4.8** | 5.0 | 291 |
| FMNIST shoes | 0.1 | **1.6** | 95.6 | 100 | 100 | 268 | 6.6 | 13.3 | 13.5 | 13.8 | 15 | 4.4 | 8.5 | **8.6** | **8.6** | 137 |
| GTS 100-rw | 8/255 | 5.1 | 13.4 | 13.4 | 13.4 | 234 | 12.6 | 18.7 | 19.0 | 19.0 | 69 | **3.8** | **7.8** | **7.8** | **7.8** | 299 |
| GTS 30-70 | 8/255 | 17.0 | 29.4 | 29.4 | 29.7 | 300 | 22.3 | 27.5 | 28.8 | 28.8 | 153 | **15.9** | **23.4** | **23.4** | 23.6 | 292 |
| **depth=4** | | | | | | | | | | | | | | | | |
| breast-cancer | 0.3 | 0.7 | 81.0 | 81.0 | 81.8 | 78 | **0.0** | 19.7 | 27.0 | 27.0 | 3 | 0.7 | **6.6** | **6.6** | **6.6** | 46 |
| diabetes | **0.05** | **22.7** | 51.3 | 55.2 | 61.7 | 18 | 26.6 | 45.5 | 46.8 | 46.8 | 1 | 27.3 | **35.7** | **35.7** | **35.7** | 9 |
| cod-rna | 0.025 | **3.4** | 37.6 | 41.6 | 47.1 | 150 | 10.9 | 24.6 | 24.8 | 24.8 | 2 | 6.9 | **21.3** | **21.3** | 21.4 | 36 |
| MNIST 1-5 | 0.3 | **0.1** | 59.1 | 90.7 | 96.0 | 72 | 1.3 | 7.1 | 9.0 | 9.5 | 5 | 0.2 | **1.3** | **1.3** | **1.4** | 126 |
| MNIST 2-6 | 0.3 | **0.4** | 89.6 | 89.6 | 100 | 79 | 2.3 | 15.1 | 15.1 | 15.9 | 6 | 0.7 | **3.8** | **3.8** | 4.1 | 88 |
| FMNIST shoes | 0.1 | **1.7** | 84.0 | 99.8 | 99.9 | 117 | 5.5 | 13.2 | 14.1 | 14.2 | 12 | 3.6 | **7.7** | 8.0 | 8.1 | 128 |
| GTS 100-rw | 8/255 | **0.9** | 5.8 | 6.0 | 6.1 | 148 | 1.0 | 5.7 | 8.4 | 8.4 | 40 | 2.6 | **4.7** | **4.7** | **4.7** | 105 |
| GTS 30-70 | 8/255 | 14.2 | 31.1 | 31.4 | 32.6 | 148 | 16.2 | 24.7 | 26.7 | 26.8 | 26 | **13.8** | **20.9** | **20.9** | 21.4 | 148 |
| **depth=8** | | | | | | | | | | | | | | | | |
| breast-cancer | 0.3 | **0.7** | 83.9 | 84.7 | 84.7 | 54 | **0.7** | 13.1 | 19.7 | 19.7 | 3 | **0.7** | **8.8** | **8.8** | **8.8** | 1 |
| diabetes | 0.05 | **22.1** | 68.8 | 83.1 | 91.6 | 27 | 29.9 | 73.4 | 77.9 | 77.9 | 1 | 27.3 | **35.7** | **35.7** | **35.7** | 2 |
| cod-rna | 0.025 | **3.2** | 38.9 | 49.0 | 61.3 | 72 | 5.6 | 28.9 | 30.8 | 31.8 | 2 | 6.6 | **21.0** | **21.1** | **21.1** | 5 |
| MNIST 1-5 | 0.3 | 0.4 | 86.6 | 92.6 | 94.5 | 28 | 1.0 | 7.2 | 10.5 | 11.4 | 5 | **0.2** | **1.0** | 1.2 | 1.4 | 60 |
| MNIST 2-6 | 0.3 | **0.4** | 78.1 | 95.1 | 99.9 | 61 | 0.8 | 9.3 | 11.7 | 12.1 | 7 | **0.4** | **2.7** | 3.0 | 3.3 | 72 |
| FMNIST shoes | 0.1 | **1.8** | 80.2 | 99.9 | 100 | 64 | 4.5 | 14.5 | 16.5 | 16.6 | 7 | 3.3 | **7.4** | 8.3 | 8.3 | 12 |
| GTS 100-rw | 8/255 | 8.7 | 19.6 | 19.7 | 20.8 | 38 | **0.9** | **6.1** | 13.3 | 13.5 | 32 | 6.0 | 10.5 | **10.6** | 11.3 | 25 |
| GTS 30-70 | 8/255 | 15.4 | 39.6 | 40.0 | 40.9 | 39 | 14.3 | 23.2 | 25.5 | 25.8 | 21 | **11.9** | **21.0** | **21.1** | 22.0 | 63 |

**Our robust boosted trees of depth 4**

**Our robust boosted trees of depth 8**

**Figure 4:** Robustness vs test error trade-off of our robust boosted trees. We can observe that robustness often comes with a loss in test error depending on the particular value of $\epsilon$. However, for FMNIST shoes, there exists a range of $\epsilon$ when robust training helps to slightly improve test error.

**Figure 5:** Feature importance of different boosted tree models on breast-cancer dataset based on the number of splits made at a particular pixel.

**Figure 6:** Feature importance of different boosted tree models on MNIST 1-5 and MNIST 2-6 based on the number of splits made at a particular pixel.

### G.6 Distribution of splitting thresholds

In Figures 8, 9, 10, we plot the distibutions of the splitting thresholds $b$ for the three boosted tree models of depth 4 on breast-cancer, MNIST 1-5, MNIST 2-6, GTS 100-rw, and GTS 30-70 datasets reported in Table 2. We can observe that our robust models on breast-cancer tend to select splits away from 0 and 1. On MNIST 1-5 and MNIST 2-6 the distributions for the normal and robust models are completely different – almost all splits for the normal model are very close to 0 and 1, while the splits for the robust model are mostly in the range between 0.3 and 0.7. This is reasonable given that more than 80% pixels of MNIST are either 0 or 1, and the considered $l_\infty$-perturbations are within $\epsilon = 0.3$. And since the normal model splits arbitrarily close to 0 or 1, this suggests that its decisions might be easily flipped if the adversary is allowed to change them within $\epsilon$. We also note that adversarially trained models have a distribution of the splitting thresholds that resembles the distribution for our models, however there are still quite many non-robust splits around 0 and 1. This again emphasizes the importance of solving the robust optimization problem properly. On GTS 100-rw and GTS 30-70 we can see that the distribution of thresholds for the robust model differs from the normal and adversarially trained models. It is interesting to note that there are no splits too close to one.

**Figure 7:** Feature importance of different boosted tree models on GTS 100-rw and GTS 30-70 based on the number of splits made at a particular pixel.

**Figure 8:** The distribution of the splitting thresholds for boosted tree models trained on breast-cancer dataset. We can observe that the choice of splitting thresholds is different for the robust model, in particular it does not have splits larger than at 1 - $\epsilon$ ($\epsilon = 0.3$).

**Figure 9:** The distribution of the splitting thresholds for boosted tree models trained on MNIST 1-5 and MNIST 2-6. We can observe that the robust model almost always select splits in the range between 0.3 and 0.7, which is reasonable according to $l_\infty$-perturbations within $\epsilon = 0.3$. At the same time, the normal model splits arbitrarily close to 0 or 1, which suggests that its decisions might be easily flipped by the adversary.

**Figure 10:** The distribution of the splitting thresholds for boosted tree models trained on GTS 100-rw and GTS 30-70. We can observe that the robust model often selects splits in the range between 8/255 ($\approx 0.031$) and 1 - 8/255 ($\approx 0.969$), which is reasonable according to $l_\infty$-perturbations within $\epsilon = 8/255$.

### G.7 Adversarial examples for boosted stumps and trees

**Exact adversarial examples for boosted stumps:**  In Section 3.1, we described how we can efficiently obtain provably minimal (exact) adversarial examples for boosted stumps. We show them for MNIST 1-5 and MNIST 2-6 datasets in Figure 11. We show the size of $l_\infty$-perturbation needed to flip the class in the title of each image. First, we can observe that $l_\infty$-perturbations are sparse which is due to the fact that we modify only the pixels that influence particular decision stumps that contribute to minimization of (4). The main observation is that the perturbations for normal models are extremely small, while for robust models they are much larger in terms of the $l_\infty$-norm. In particular, they have usually $\|\delta\|_\infty$ slightly larger than $0.3$ which makes sense since the $\epsilon$ that we used during training was equal to $0.3$. Moreover, for robust models, the perturbations are situated at the locations where one can expect pixels of the opposite classes.

**Adversarial examples for boosted trees:**  Adversarial examples for different boosted tree models are obtained via the binary search applied on top of the cube attack. We show the resulting images in Figure 12 for MNIST 1-5 and MNIST 2-6, and in Figure 13 for GTS 100-rw and GTS 30-70. We note that qualitatively the adversarial examples for boosted trees are very similar to the exact adversarial examples for boosted stumps. Except that for a few images the perturbation is larger in $l_\infty$-norm and affects more pixels. This might be an artifact of how the cube attack works, although for visualization purposes we remove the perturbations from the features that do not affect any splits. For GTS 100-rw and GTS 30-70, we see that the changes that flip the class are often quite small even for our robust models which is due to the fact that we used a small $\epsilon$ during training ($8/255$) which is much lower than the $\epsilon$ for MNIST 1-5 or MNIST 2-6. We can see noticeable changes mostly for the images that have a natural contrast level. For low-contrast images the changes are harder to spot, but they are still present at the locations shown on the heatmaps from Figure 7.

Overall, we can conclude that for boosted stumps and trees the presented adversarial examples do not show perceptual interpolations between classes like robust neural networks [64], but this we cannot expect from such simple classifiers. What is more important in the context of stumps and trees is rather the idea of instance-based explanations that can help to get more insights into how the model makes its decisions.

**Figure 11:** Exact adversarial examples for boosted stumps trained on MNIST 1-5 and MNIST 2-6 datasets. We show the size of $l_\infty$-perturbation needed to flip the class in the title of each image. We can observe that perturbations for normal models are extremely small or even imperceptible, while for robust models they are much larger in $l_\infty$-norm and situated at the locations where one can expect pixels of the opposite classes.

| Normal trees | Adv. trained trees | Our robust trees | Normal trees | Adv. trained trees | Our robust trees |
|---|---|---|---|---|---|
| $\|\delta\|_\infty=0.375$ | $\|\delta\|_\infty=0.305$ | $\|\delta\|_\infty=0.490$ | $\|\delta\|_\infty=0.020$ | $\|\delta\|_\infty=0.297$ | $\|\delta\|_\infty=0.303$ |
| $\|\delta\|_\infty=0.008$ | $\|\delta\|_\infty=0.305$ | $\|\delta\|_\infty=0.303$ | $\|\delta\|_\infty=0.020$ | $\|\delta\|_\infty=0.500$ | $\|\delta\|_\infty=0.303$ |
| $\|\delta\|_\infty=0.375$ | $\|\delta\|_\infty=0.309$ | $\|\delta\|_\infty=0.496$ | $\|\delta\|_\infty=0.061$ | $\|\delta\|_\infty=0.500$ | $\|\delta\|_\infty=0.314$ |
| $\|\delta\|_\infty=0.504$ | $\|\delta\|_\infty=0.500$ | $\|\delta\|_\infty=0.539$ | $\|\delta\|_\infty=0.020$ | $\|\delta\|_\infty=0.500$ | $\|\delta\|_\infty=0.305$ |
| $\|\delta\|_\infty=0.004$ | $\|\delta\|_\infty=0.305$ | $\|\delta\|_\infty=0.303$ | $\|\delta\|_\infty=0.020$ | $\|\delta\|_\infty=0.500$ | $\|\delta\|_\infty=0.303$ |
| $\|\delta\|_\infty=0.004$ | $\|\delta\|_\infty=0.305$ | $\|\delta\|_\infty=0.303$ | $\|\delta\|_\infty=0.037$ | $\|\delta\|_\infty=0.375$ | $\|\delta\|_\infty=0.303$ |
| $\|\delta\|_\infty=0.004$ | $\|\delta\|_\infty=0.375$ | $\|\delta\|_\infty=0.303$ | $\|\delta\|_\infty=0.012$ | $\|\delta\|_\infty=0.312$ | $\|\delta\|_\infty=0.303$ |
| $\|\delta\|_\infty=0.008$ | $\|\delta\|_\infty=0.305$ | $\|\delta\|_\infty=0.303$ | $\|\delta\|_\infty=0.059$ | $\|\delta\|_\infty=0.305$ | $\|\delta\|_\infty=0.305$ |

**Figure 12:** Adversarial examples for boosted trees trained on MNIST 1-5 and MNIST 2-6 datasets. We show the size of $l_\infty$-perturbation needed to flip the class in the title of each image. We can observe that perturbations for normal models are extremely small or even imperceptible, while for robust models they are much larger in $l_\infty$-norm and situated at the locations where one can expect pixels of the opposite classes.

| Normal trees | Adv. trained trees | Our robust trees | Normal trees | Adv. trained trees | Our robust trees |
|:---:|:---:|:---:|:---:|:---:|:---:|
| $\|\delta\|_\infty=0.533$ | $\|\delta\|_\infty=0.500$ | $\|\delta\|_\infty=0.484$ | $\|\delta\|_\infty=0.047$ | $\|\delta\|_\infty=0.076$ | $\|\delta\|_\infty=0.125$ |
| $\|\delta\|_\infty=0.250$ | $\|\delta\|_\infty=0.219$ | $\|\delta\|_\infty=0.227$ | $\|\delta\|_\infty=0.250$ | $\|\delta\|_\infty=0.252$ | $\|\delta\|_\infty=0.270$ |
| $\|\delta\|_\infty=0.078$ | $\|\delta\|_\infty=0.063$ | $\|\delta\|_\infty=0.094$ | $\|\delta\|_\infty=0.084$ | $\|\delta\|_\infty=0.047$ | $\|\delta\|_\infty=0.078$ |
| $\|\delta\|_\infty=0.102$ | $\|\delta\|_\infty=0.094$ | $\|\delta\|_\infty=0.125$ | $\|\delta\|_\infty=0.125$ | $\|\delta\|_\infty=0.094$ | $\|\delta\|_\infty=0.062$ |
| $\|\delta\|_\infty=0.125$ | $\|\delta\|_\infty=0.043$ | $\|\delta\|_\infty=0.250$ | $\|\delta\|_\infty=0.227$ | $\|\delta\|_\infty=0.141$ | $\|\delta\|_\infty=0.191$ |
| $\|\delta\|_\infty=0.094$ | $\|\delta\|_\infty=0.084$ | $\|\delta\|_\infty=0.062$ | $\|\delta\|_\infty=0.324$ | $\|\delta\|_\infty=0.281$ | $\|\delta\|_\infty=0.326$ |
| $\|\delta\|_\infty=0.125$ | $\|\delta\|_\infty=0.082$ | $\|\delta\|_\infty=0.094$ | $\|\delta\|_\infty=0.250$ | $\|\delta\|_\infty=0.109$ | $\|\delta\|_\infty=0.250$ |
| $\|\delta\|_\infty=0.125$ | $\|\delta\|_\infty=0.047$ | $\|\delta\|_\infty=0.125$ | $\|\delta\|_\infty=0.070$ | $\|\delta\|_\infty=0.117$ | $\|\delta\|_\infty=0.094$ |

**Figure 13:** Adversarial examples for boosted trees trained on GTS 30-70 and GTS 100-rw datasets. We show the size of $l_\infty$-perturbation needed to flip the class in the title of each image. We see that the changes are often quite small even for our robust models which is due to the fact that we used a small $\epsilon$ during training $(8/255)$ which is much lower than the $\epsilon$ for MNIST 1-5 or MNIST 2-6.