[Reviews · NeurIPS 2019]

Reviewer 1



--- Summary --- Contributions: This paper studies the adversarial example problem for ensembles of decision trees and decision stumps (trees with a single internal node). As a main contribution, the authors derive an exact attack algorithm on ensembles of decision stumps for \ell_\infty perturbations. In contrast, this problem is known to be NP-Hard for trees with at least 3 internal nodes, via previous work by Kantchelian et. al., reference [22] in the present paper. The current work uses this attack to derive an optimal robust training method for stump ensembles. They also provide an approximate upper bound on the robust test error for tree ensembles, which improves upon the naive upper bounds while being efficiently computable. Quality: My main concern about the present work is a lack of detailed comparison with two very relevant prior works, which are references [7] and [22] in the present paper. For one, there is no stand-alone related work section, and comparisons to previous work are only made in passing. The bigger issue is that the experimental claims in this paper are not compared to strong enough baselines, even though previous work is relevant enough to warrant a direct comparison. Specifically, the robust decision trees in [7] should be evaluated side-by-side with the current algorithms, as they may have much better robust accuracy empirically. Similarly, the attack algorithm in [22] is quickly ruled out as not scaling to large datasets due to using a MILP formulation. However, in the case of decision stumps, the MILP may very well be solvable quickly, especially on small datasets. In fact, the previous work [7] solves the MILP exactly to attack tree ensembles over three datasets that are also used in the present paper: diabetes, cod-rna, and breast-cancer. Therefore, it would be much more meaningful to establish the exact robust test accuracy, and to compare the present robust models to the approximate robust decision trees of the previous work. Clarity: The paper is generally well-written and structured clearly. The main theoretical results are clearly and soundly justified. The experimental set-up and results are also very clear and concise. Significance: As mentioned above, it is difficult to judge the significance of the present work without a better comparison to previous work. On a different note, the focus on decision stumps is also only moderately motivated by their interpretability. The core reason that the new attack and the new robust training is tractable is that decision stumps are separable due to each stump only depending on a single coordinate (and because the perturbations are in the \ell_\infty norm). Therefore, the theoretical results in the paper are not especially deep, even though they are certainly novel and interesting. --- Review --- Pros: - The paper presents interesting ideas on how to construct provably robust decision stumps, which was previously unknown in the literature. - This follows from a new exact attack algorithm for stump ensembles, which uses a simple but elegant observation about the separability for \ell_\infty perturbations. - The authors also present an approximate certificate on the robust accuracy of tree ensembles, which was also previously unknown, and it utilizes the new result for stumps (applying a greedy, coordinate-based approach). - The results are clearly described. The writing is well done. Cons: - Some more motivation for focusing on decision stumps would be nice. They are quite a restricted class of tree ensembles. Why is it important to understand adversarial robustness for stump ensembles? Alternatively, can the ideas in the present paper be extended to a richer class of boosted models? - Many of the results should be explained in the context of the previous work [7], which achieves quite strong results empirically in the same setting as the current work. The work [7] is also on robust decision trees, and it appeared in ICML 2019. Therefore, an experimental comparison with [7] is warranted but missing from this submission. - In particular the present authors say "While the approach of [7] leads to tree ensembles with improved empirical and certified robustness, they have no approximation guarantee on the robust loss or robust error during training time. In contrast we show how to derive an upper bound on the robust loss for tree ensembles based on our results for an ensemble of decision stumps and we show how that upper bound can be minimized during training." So the natural question is how do the robust decision trees in the previous work [7] compare with the results in the present work? Is one or the other more robust/accurate on some datasets, so is there a trade-off for certifiability? - It should be possible to run the MILP in [22] for at least three datasets in the present work, which would lead to a better comparison of the exact versus approximate robust accuracy of tree ensembles (as is done in the previous work [7] for the same datasets). - A standard approach to construct robust classifiers is adversarial training. In particular, [22] shows that performing many rounds of adversarial training in the same norm as the attack may be an effective way to create robust tree ensembles (although [22] only evaluates \ell_0 distance). The present work does not compare against any adversarial training approaches, which I believe is a serious shortcoming and should be addressed as it may be a competitive baseline. After author feedback: I will increase my score to a 6, given the thoroughness of the feedback. The new experiments and comparison to [7] and [22] are enlightening and better showcase the improvements of the present work. Please include much more details about the importance of decision stumps (e.g. interpretability) in the final version, as well as the experiments and discussion from the feedback.

Reviewer 2



The authors proposed a provable solution to adversarial training of decision stumps and tree ensembles and also exact and upper bounds of the "robust test error". This problem is hard in general for arbitrary complicated classifiers, because the inner maximization in adversarial training is a non-concave problem in general. But the authors show that the inner maximization is concave in case of boosted decision stumps by exploiting the fact that the objective function is piecewise constant and one could perform maximization by sorting the "boundary points" in the piecewise constant function. For boosted decision trees, one could rely on upper bounds on the robust error, which are found to be tight empirically. Empirical results show that the model is indeed provably robust for reasonably sized perturbation (l_inf norm is considered). The paper overall reads pretty well and the idea sounds interesting. But I have some concerns regarding the evaluations: 1) It would make sense to compare the method against [7] at least for the datasets that are not too big. This could better highlight advantages of using the proposed method over [7]. 2) As another baseline, it would be insightful to compare the results against other classifiers that are adversarially trained (that use approximations in the inner optimization), for example using Projected Gradient Ascent. This could highlight the importance of how solving the inner optimization accurately could lead to better robustness. 3) The training computational complexity is O(n^2 T log T), which means that O(n) passes over the entire dataset may be needed to train the classifier. This may limit practical use of the proposed method for huge datasets, where O(n^2) could be prohibitive. Is there a way to bring the complexity down to less than O(n^2) running time?

Reviewer 3



This paper seeks to bridge two areas - the use of boosting and tree-like models as well as robustness in machine learning. They show that for such models, it is possible to derive the optimal robust loss (for decision stumps) or an upper bound on the robust loss (for decision trees) and also propose a training procedure for each type of model that shows empirical improvements in robust accuracy. The mathematical analysis of the proposed loss computation algorithm and the related training algorithm is solid and seems correct. The authors do make some simplifications such as assuming the loss function is an exponential loss, but claim (without proof) that the analysis also applies to other strictly monotonically decreasing loss functions such as logistic loss. The math is explained well in a step by step manner. Still, I would have found it clearer if it had an “algorithm” section that lists out the steps of the algorithm described in Section 3.2 in bullet point form. This would also make the runtime complexity analysis more clear. On the empirical side, the results show an improvement in robust test accuracy by using the proposed algorithm. The improvement is clear and it is done over a range of different datasets. However, I would focus more on the interpretability of the resulting models. In particular, I liked the discussion in Figure 2 analyzing exactly how robust boosted stumps on MNIST 2-6 split differently from regular boosted stumps. Interpretability is one of the main advantages of these types of models, so I would be more interested in seeing even more analysis on this front. It is okay to place the analysis in the Appendix, but I would be particularly interested in seeing if the robust and interpretable trees say anything interesting about the health-related datasets. Otherwise, if you ignore the interpretability of such models, the robustness result on MNIST is easily surpassed by deep neural network models. Additionally, on the empirical side, I believe it is worth comparing results directly to [7] whenever possible, as the work of [7] seems quite similar. While there are differences in the approach taken to train such robust models and obtain robust loss guarantees, both approaches do obtain robust loss guarantees by the end. Thus, having a side-by-side comparison of the results whenever a fair comparison is possible seems important to include (again, having this in the Appendix is okay too). Finally, a small interesting note the authors mention is that a simple sampling method tends to result in a close approximation to the true robust test error for boosted trees for small datasets like the ones used in the paper. Thus, it may be worth having a training method based on sampling to compute the robust loss as another baseline algorithm to compare to. This is akin to having a (possibly weak) adversarial training procedure as a baseline for provable robustness in deep neural networks. Overall, the paper has room for improvement (especially on the empirical analysis side), but addresses an interesting question in a mathematically sound way. Small comments Line 9 - “Up to our knowledge” -> “To the best of our knowledge” Line 71 - “and allows to see” -> “and allows us to see” Line 77 - what is N_t? Line 99-100 - “Our goal … are provable guarantees” -> “Our goal … is provable guarantees” Line 105 - “in original Adaboost” -> “in the original Adaboost” Line 105 - “successfully in object detection” -> “successfully used in object detection” Line 135 - It maybe worth mentioning somewhere closer to the beginning of section 3.2 that you’ll first describe what to do for a specific coordinate j, then choose which coordinates j you will use later (you discuss this in lines 163-165). Again, this could also be improved with an “algorithm” section that has an algorithm with bullet points. Line 243 - “to judge about robustness” -> “to judge robustness” Line 264 - “which allows to” -> “which allows us to” *** After Author Response *** I am satisfied with the additional experimental results presented in the author response. Again, as discussed previously, I hope the authors do indeed have a longer discussion on interpretability as that is probably one of the main advantages of using decision trees/stumps. I will keep my score as an accept.

[Author Response · NeurIPS 2019]

We thank the reviewers for the constructive feedback and detailed comments which we integrate in the final version.

**R1/R2/R3:** *"comparison with [7] for boosted decision trees and to neural networks"*
At submission time no code was available for [7] which is why we did not compare to them. Please note that we
optimize directly an upper bound on the adversarial loss whereas this is only approximately true for [7]. In Table 1 we
compare our provably robust boosted trees to [7] (same setting for [7] as ours: we fit boosted depth 4 trees with 80% of
the training data and use the rest as validation set for model selection). For [7] we use the exact robust test error (RTE)
[22] for model selection, whereas for us we use our upper bound on RTE (URTE). For [7] we use a coarser grid for
large number of iterations as RTE is expensive to evaluate. We see that our URTE is for 6 out of 7 datasets smaller than
their RTE sometimes with large margin e.g. on diabetes. Our better URTE comes at the price of worse test error but this
is a well-known phenomena for neural networks, that methods enforcing better RTE suffer in test error. Our LRTE
values improved as we have come up with a new attack scheme - now LRTE and URTE are tight.

Table 1: Comparison of the boosted trees of [7] to the results of our boosted trees reported in the paper. The shown time is for boosted trees of [7] the computation of the RTE for the final model with the MILP of [22] (adapted to a feasibility problem for existence of an adv. example within $l_\infty$-ball) and for URTE with our algorithm. All numbers are for the full test set.

| Dataset | $l_\infty\ \epsilon$ | Chen et al [7], depth=4 | | | | Our provably robust trees, depth=4 | | | | |
|---|---|---|---|---|---|---|---|---|---|---|
| | | TE | RTE | # trees | Time RTE [22] | TE | LRTE | URTE | # trees | Time URTE (ours) |
| breast-cancer | 0.3 | 0.7 | 13.1 | 8 | 5.3s | 2.9 | 10.2 | **10.2** | 1 | **0.1ms** |
| diabetes | 0.05 | 22.1 | 40.3 | 5 | 4.0s | 28.6 | 33.1 | **33.1** | 3 | **0.8ms** |
| cod-rna | 0.025 | 10.2 | 24.2 | 20 | 1.4h | 8.3 | 23.2 | **23.2** | 14 | **0.5s** |
| MNIST 2-6 | 0.3 | 0.5 | 6.9 | 1000 | 2.0m | 0.7 | 4.8 | **5.0** | 47 | **0.2s** |
| FMNIST shoes | 0.1 | 3.1 | 13.2 | 20 | 58.3s | 4.7 | 10.5 | **10.5** | 8 | **0.1s** |
| GTS 100-rw | 8/255 | 1.5 | **9.7** | 20 | 35.9s | 4.7 | 10.1 | 10.1 | 11 | **0.2s** |
| GTS 30-70 | 8/255 | 11.5 | 28.8 | 20 | 23.4s | 14.9 | 27.2 | **27.2** | 14 | **0.4s** |
| MNIST | 0.3 | 2.0 | 31.2 | 200 | 2.7h | 4.8 | 14.6 | **18.5** | 55 | **2.5m** |
| FMNIST | 0.1 | 14.4 | 65.1 | 200 | 3.8h | 15.3 | 23.5 | **25.4** | 25 | **1.2m** |

We extended our approach to multi-class using one-vs-all. We fitted tree ensembles of depth 14. For MNIST with
$\epsilon = 0.3$ we get a URTE of 18.5% versus 31.2% for [7]. Our URTE is better than that reported for neural networks
(NNs) (33.6% [45], 19.3% [47]) and only the very recent [17] improved this to 8.1%. For FMNIST we get 25.4%
URTE vs 65.1% RTE for [7] whereas NNs achieve 30.7% URTE [10] (with 26.6% LRTE) so that our tree ensemble is
more robust. **This shows that regarding provable robustness tree ensembles can be competitive with NNs.**

**R1/R2/R3:** *"comparison to adversarial training (AT)"*
We tried AT as in [22] and obtained much worse robustness than ours. Different from the $l_0$-attack of [22] for an
$l_\infty$-attack all features are perturbed and that leads to suboptimal initial splits from which the ensemble does not recover.
We think that AT should not be used if one has a tight and scalable upper bound on the robust loss as AT provides only
a lower bound and minimization of an upper bound makes more sense than minimization of a lower bound.

**R1:** *"c) Approximate upper bounds on robustness of stump and tree ensembles".*
We want to clarify that our upper bound on the adversarial loss is not approximate, but a strict upper bound.

**R1:** *"in the case of decision stumps, the MILP may very well be solvable quickly".*
The MILP of [22] scales up to larger tree ensembles when changing it to a feasibility problem for the computation of
the RTE rather than the minimal adv. perturbation. However, our upper bound computation which is very tight (see
Table 1) is about 100x faster. For decision stumps our exact algorithm has runtime complexity $O(nT \log T)$ whereas
the MILP has no polynomial runtime guarantees and in practice the MILP is several times slower. We can't compare
the running time directly as we need time to transfer our tree ensembles into the code of [7] to access the MILP [22].

**R1:** *"the theoretical results in the paper are not especially deep, even though they are certainly novel and interesting"*
Scalable provably robust training need not be complicated. IBP [17] the state-of-the-art method for provably robust
NNs is based on "simple" interval arithmetics, and theoretically less involved than [32, 44] which are hard to scale.

**R1:** *"Some more motivation for focusing on decision stumps would be nice."*
For simple data sets boosted stumps are sufficient and more interpretable than boosted trees. In terms of RTE our exact
decision stumps outperform our robust tree ensemble on diabetes and cod-rna. Apart from linear models this is up to
our knowledge the first scalable, exact algorithm for the minimization of the robust loss. See also [15] for an example
where boosted stumps have better generalization properties than boosted trees.

**R2:** *"Is there a way to bring down the training computational complexity down from $O(n^2)$"*
We investigate if it can be improved to $O(n \log n)$ but have not succeeded yet. One heuristic is to use a small subset of
the thresholds for large datasets. Empirically, this yields only small loss in URTE but a significant speed-up of training.

**R3:** *"... more discussion of interpretability of boosted decision trees/stumps"*
We agree that this is very interesting and will include more analysis along the lines we have done already.

[Meta-Review · NeurIPS 2019]

Thank you for your submission to NeurIPS. After the author response and discussion, the reviewers and I are in agreement that this work presents an interesting and substantial contribution to the work on provably robust adversarial learning. The extension of such methods from the typical NN setting to one of boosted decision stumps is an interesting one, and certainly worthy of publication. The author response in particular was good at addressing the points of one of the initially most negative reviewer, and it would be good to include these points into the final version.